

# New Observations of Upper Tropospheric NO₂ from TROPOMI

Eloise A. Marais[1,2], John F. Roberts[3], Robert G. Ryan[4,5], Henk Eskes[6], K. Folkert Boersma[6,7], Sungyeon Choi[8,9], Joanna Joiner[8], Nader Abuhassan[8,10], Alberto Redondas[11], Michel Grutter[12], Alexander Cede[13], Laura Gomez[14,15], Monica Navarro-Comas[14]

[1]Department of Geography, University College London, London, UK
[2]School of Physics and Astronomy, University of Leicester, Leicester, UK
[3]Centre for Landscape and Climate Research, University of Leicester, Leicester, UK
[4]School of Earth Sciences, The University of Melbourne, Melbourne, Australia
[5]ARC Centre of Excellence for Climate System Science, Sydney, Australia
[6]Satellite Observations Department, Royal Netherlands Meteorological Institute (KNMI), De Bilt, the Netherlands
[7]Meteorology and Air Quality Group, Wageningen University (WUR), Wageningen, the Netherlands
[8]NASA Goddard Space Flight Center, Greenbelt, MD, USA
[9]Science Systems and Applications, Inc., Lanham, MD, USA
[10]Joint Center for Earth Systems Technology, University of Maryland, Baltimore County, Baltimore, MD, USA
[11]Izaña Atmospheric Research Center, AEMET, Tenerife, Canary Islands, Spain
[12]Centro de Ciencias de la Atmósfera, Universidad Nacional Autónoma de México, Mexico City, Mexico
[13]LuftBlick, Fritz-Konzert-Straße 4, Innsbruck, Austria
[14]Instituto Nacional de Técnica Aeroespacial (INTA), Área de Investigación e Instrumentación Atmosférica, Ctra Ajalvir km4, 28850, Torrejón de Ardoz, Madrid, Spain
[15]Groupe de Spectrométrie Moléculaire et Atmosphérique, URM CNRS 7331, UFR Sciences Exactes et Naturelles, Moulin de la Housse, BP 1039, 51687 Reims CEDEX 2, France

*Correspondence to*: Eloise A. Marais (e.marais@ucl.ac.uk)

**Abstract.** Nitrogen oxides ($NO_x \equiv NO + NO_2$) in the $NO_x$-limited upper troposphere (UT) are long-lived and so have a large influence on the oxidizing capacity of the troposphere and formation of the greenhouse gas ozone. Models misrepresent $NO_x$ in the UT and observations to address deficiencies in models are sparse. Here we obtain a year of near-global seasonal mean mixing ratios of $NO_2$ in the UT (450-180 hPa) at $1° \times 1°$ by applying cloud-slicing to partial columns of $NO_2$ from TROPOMI. This follows refinement of the cloud-slicing algorithm with synthetic partial columns from the GEOS-Chem chemical transport model. We find that synthetic cloud-sliced UT $NO_2$ are spatially consistent (R = 0.64) with UT $NO_2$ calculated across the same cloud pressure range and scenes as are cloud-sliced ("true" UT $NO_2$), but the cloud-sliced UT $NO_2$ is 11-22% more than the "true" all-sky seasonal mean. The largest contributors to differences between synthetic cloud-sliced and "true" UT $NO_2$ are target resolution of the cloud-sliced product and uniformity of overlying stratospheric $NO_2$. TROPOMI, prior to cloud-slicing, is corrected for a 13% underestimate in stratospheric $NO_2$ variance and a 50% overestimate in free tropospheric $NO_2$ determined by comparison to Pandora total columns at high-altitude sites in Mauna Loa, Izaña and Altzomoni, and MAX-DOAS and Pandora tropospheric columns at Izaña. Two cloud-sliced seasonal mean UT $NO_2$ products for June 2019 to May 2020 are retrieved from corrected TROPOMI total columns using distinct TROPOMI cloud products that assume clouds are reflective boundaries (FRESCO-S) or water droplet layers (ROCINN-CAL). TROPOMI UT $NO_2$ typically ranges from 20-30 pptv over remote oceans to >80 pptv over locations with intense seasonal lightning. Spatial coverage is mostly in the tropics





and subtropics with FRESCO-S and extends to the midlatitudes and polar regions with ROCINN-CAL, due to its greater abundance of optically thick clouds and wider cloud top altitude range. TROPOMI UT NO$_2$ seasonal means are spatially

consistent (R = 0.6-0.8) with an existing coarser spatial resolution (5° latitude × 8° longitude) UT NO$_2$ product from the Ozone Monitoring Instrument (OMI). UT NO$_2$ from TROPOMI is 12-26 pptv more than that from OMI due to increase in NO$_2$ with altitude from the OMI pressure ceiling (280 hPa) to that for TROPOMI (180 hPa), but possibly also systematic altitude differences between the TROPOMI and OMI cloud products. The TROPOMI UT NO$_2$ product offers potential to evaluate and improve representation of UT NO$_x$ in models and supplement aircraft observations that are sporadic and susceptible to large

biases in the UT.

## 1 Introduction

Nitrogen oxides (NO$_x$ ≡ NO+NO$_2$) in the upper troposphere (UT; ~8-12 km) influence the oxidizing capacity of the atmosphere and global climate, as the formation and radiative forcing of tropospheric ozone are most efficient in the NO$_x$-limited UT (Mickley et al., 1999; Bradshaw et al., 2000; Dahlmann et al., 2011; Worden et al., 2011). Sources of NO$_x$ to the UT include

local emissions from lightning and cruising altitude aircraft, deep convective uplift of surface pollution, downwelling from the stratosphere, long-range transport, and chemical recycling of NO$_x$ from stable reservoir compounds (Ehhalt et al., 1992; Lamarque et al., 1996; Schumann, 1997; Jaeglé et al., 1998; Bradshaw et al., 2000; Bertram et al., 2007). The lifetime of NO$_x$ in the UT varies from a few hours to a few days depending on availability of hydrogen oxides (HO$_x$ ≡ OH + HO$_2$) and peroxy radicals (RO$_2$) to convert NO$_x$ to reservoir compounds (Jaeglé et al., 1998; Bradshaw et al., 2000; Nault et al., 2016).


Current understanding of UT NO$_x$ is erroneous, as demonstrated by misrepresentation in chemical transport models (CTMs) of the vertical distribution, relative abundance (ratios of NO-to-NO$_2$), and absolute magnitude of UT NO$_x$ when compared to in situ measurements from research aircraft (Boersma et al., 2011; Travis et al., 2016; Silvern et al., 2018). Models are used to determine the contribution of ozone to anthropogenic climate change in the absence of reliable historical measurements

(Pavelin et al., 1999). Models also provide prior information about the vertical distribution of NO$_2$ for retrieval of vertical column densities of NO$_2$ from space-based UV-visible instruments. Errors in these retrievals are particularly vulnerable to biases in modelled UT NO$_2$, due to greater sensitivity of space-based observations to the UT than the middle and lower troposphere (Travis et al., 2016; Silvern et al., 2019). This impedes accurate top-down inference of air quality variability, surface concentrations and precursor emissions (Stavrakou et al., 2013; Silvern et al., 2019). Models include heavily

parameterized representation of lighting (Tost et al., 2007; Allen et al., 2010; Ott et al., 2010; Murray et al., 2012; Murray et al., 2013), the largest global influencer of NO$_x$ in the UT (Bradshaw et al., 2000; Marais et al., 2018), and may misrepresent the reaction kinetics and physical processing of NO$_x$ for the cold, low-pressure conditions of the UT (Chang et al., 2011; Henderson et al., 2011; 2012; Stavrakou et al., 2013; Amedro et al., 2019).





Observations that have been used to better understand UT NO$_x$ are mostly limited to research and commercial aircraft campaigns. For research aircraft, the record of observations in the UT since the early 1990s have been sustained almost exclusively by the NASA DC8 plane, with recent contributions from the German High Altitude and Long Range Research Aircraft (HALO) (Wendisch et al., 2016). There are also commercial aircraft campaigns, but these are prevalent over heavily trafficked flight corridors and are often in the stratosphere at cruising altitude (Thomas et al., 2015; Stratmann et al., 2016). In

situ measurements of NO$_2$ in the UT can also be biased by interference from NO$_x$ reservoir compounds that thermally decompose to NO$_2$ in the instrument inlet (Browne et al., 2011; Reed et al., 2016). Standard remote sensing products of NO$_2$ from space-based nadir- and limb-viewing instruments provide global coverage, but either as a single piece of vertical information in the troposphere in the nadir as tropospheric column densities (Levelt et al., 2018) or as vertically resolved NO$_2$ in the limb limited to NO$_2$ abundances above the tropopause (Newchurch et al., 1996; Sioris et al., 2004; Brohede et al., 2007;

Jones et al., 2012).

Near-global research products of seasonal mean vertically resolved tropospheric NO$_2$ have been retrieved by applying the cloud-slicing technique to partial columns of NO$_2$ from the space-based Ozone Monitoring Instrument (OMI) (Choi et al., 2014; Belmonte-Rivas et al., 2015). Cloud-slicing involves regressing clusters of partial NO$_2$ columns above optically thick

clouds against corresponding cloud top pressures. The resultant regression slopes are converted to NO$_2$ mixing ratios that represent average NO$_2$ across the cloud top altitude range (Ziemke et al., 2001). The advantages of cloud-slicing include enhanced signal over bright optically thick clouds (van der A et al., 2020) and removal of the dry stratosphere due to lack of clouds there. Near-global multiyear (2005-2007) seasonal means of UT NO$_2$ from cloud-sliced OMI partial columns have been shown to reproduce the spatial variability of UT NO$_2$ measured with bias-corrected NASA DC8 aircraft measurements of NO$_2$

over North America, though at very coarse scales (seasonal, 32° × 20°) (Marais et al., 2018). Even so, the OMI product confirms the dominant global influence of lightning on UT NO$_x$ and provides global constraints on lightning NO$_x$ production rates (280 ± 80 moles NO$_x$ per lightning flash) and annual lightning NO$_x$ emissions (5.9 ± 1.7 Tg N) (Marais et al., 2018). OMI pixels are at relatively coarse resolution (13 km × 24 km at nadir) and there is substantial data loss after 2007 due to the so-called row anomaly (Schenkeveld et al., 2017; Torres et al., 2018). The recently launched (October 2017) TROPOMI

instrument on the Sentinel-5P satellite has the same spatial coverage as pre-row-anomaly OMI (swath width of 2600 km), but with a finer nadir pixel resolution of 7.2 km × 3.5 km (along track × across track) until 5 August 2019, further refined thereafter to 5.6 km × 3.5 km (Argyrouli et al., 2019). This offers better cloud-resolving capability and greater data pixel density than OMI with potential to retrieve finer resolution NO$_2$ in the UT.

Here we refine and test the cloud-slicing retrieval using synthetic partial NO$_2$ columns from the GEOS-Chem CTM before retrieving UT NO$_2$ from TROPOMI partial NO$_2$ columns with cloud information from two distinct TROPOMI cloud products. Application of cloud-slicing to TROPOMI follows evaluation of TROPOMI total, stratospheric and tropospheric columns with



ground-based measurements of NO₂ from Pandora and Multi-axis differential optical absorption spectroscopy (MAX-DOAS) at free tropospheric monitoring sites. We also evaluate TROPOMI UT NO₂ with the OMI UT NO₂ product.

## 2 Cloud-slicing of GEOS-Chem synthetic partial columns

Targeting cloudy scenes could yield representation errors in NO₂ mixing ratios in the UT, due to the influence of clouds on NOₓ photochemistry (Holmes et al., 2019), large enhancements in NOₓ from lightning and convective uplift of surface pollution that accompany cloud formation (Price and Rind, 1992; Bertram et al., 2007), and low sampling frequency due to strict data filtering (Choi et al., 2014). We test the ability of the cloud-slicing technique to return accurate, representative mixing ratios of NO₂ in the UT by applying this technique to synthetic partial columns from GEOS-Chem. The "true" NO₂ used to evaluate cloud-sliced NO₂ is obtained by averaging NO₂ across the same vertical range as the cloud-sliced NO₂ for the same cloudy model grid squares as are cloud-sliced ("true" cloudy UT NO₂) and for all clear and cloudy model grid squares ("true" all-sky UT NO₂).

Synthetic NO₂ are from GEOS-Chem version 12.1.0 (https://doi.org/10.5281/zenodo.1553349; last accessed 10 August 2019) simulated at a horizontal resolution of 0.25° × 0.3125° (latitude × longitude) extending over 47 vertical layers from the surface to 0.01 hPa for the nested domains available in version 12.1.0. These include North America (9.75-60°N, 130-60°W), western Europe (30-70°N, 15°W-61.25°E), and Southeast Asia (15-55°N, 70-140°E). Dynamic (3-hourly) boundary conditions are from a coarse resolution (4° × 5°) global GEOS-Chem simulation. The model is driven with NASA GEOS-FP assimilated meteorology and includes comprehensive emission inventories from anthropogenic and natural sources. These include local emissions of NOₓ in the UT from lightning as described by Murray et al. (2012) and from aircraft using the Aviation Emissions Inventory Code (AEIC) inventory detailed in Stettler et al. (2011). The model is simulated in boreal summer (June-August) when variability in UT NOₓ in all nested domains is dominated by lightning (Marais et al., 2018). The model is sampled daily at 12h00-15h00 local time (LT) to be consistent with the TROPOMI overpass time (13h30 LT). Two years (2016 and 2017) are simulated to increase data density. The model years predate TROPOMI, but this has no bearing on assessment of the cloud-slicing technique.

The cloud-slicing approach we apply to synthetic partial columns above synthetic clouds to estimate seasonal means of UT NO₂ is the same as will be applied to TROPOMI, so model variables are only used if these are also available in or can be derived from publicly available TROPOMI data products. GEOS-Chem daily partial NO₂ column densities (stratosphere + partial troposphere) and the corresponding GEOS-FP cloud top pressures at 450-180 hPa and 0.25° × 0.3125° are gathered into grid squares of the target resolution of 4° × 5°. These are then screened to remove clusters with non-uniform GEOS-Chem stratospheric NO₂ (stratospheric column NO₂ relative standard deviation > 0.02) using GEOS-FP thermal tropopause heights to determine the vertical extent of the stratosphere in the model. For a target resolution of 4° × 5° clusters of as many as 256



$0.25° \times 0.3125°$ partial columns are likely, so we increase the number of possible cloud-sliced $NO_2$ retrievals by subdividing clusters of at least 100 partial $NO_2$ columns into scenes of at least 40. This doubles the number of cloud-sliced $NO_2$ data used to obtain multiyear seasonal means. Additional filtering is applied to clusters to remove extreme $NO_2$ partial columns (partial columns falling outside the $10^{th}$ to $90^{th}$ percentile range) that have a large influence on regression of $NO_2$ partial columns against cloud top pressures, clusters with fewer than 10 partial columns after screening for extreme values, and clusters that

do not extend across a sufficiently wide altitude range (cloud top pressure range $\leq 140$ hPa and standard deviation $\leq 30$ hPa). GEOS-Chem cloud top heights are diagnosed in the model as the pressure at the top edge of the highest model layer of GEOS-FP upward moist convective mass flux.

The slope of the relationship between cloud top heights and partial columns for each cluster is estimated with reduced major

axis (RMA) regression and the error on the slope with bootstrap resampling. Additional filtering is applied to retain slopes that have low relative error (relative error on the slope $\leq 1.0$). Large local enhancements in $NO_2$ at high altitudes that lead to negative slopes and negative cloud-sliced UT $NO_2$ are diagnosed as slopes significantly less than zero (sum of slope and slope error $< 0$) and removed. The retained slopes and errors (in molecules $cm^{-2}$ $hPa^{-1}$) are converted to mixing ratios (in pptv) and outliers caused by steep slopes (UT $NO_2 > 200$ pptv) removed. A threshold of 200 pptv is used, as this far exceeds the maximum

seasonal mean UT $NO_2$ of 145 pptv in the OMI cloud-sliced UT $NO_2$ product (Marais et al., 2018). We find though that only 3 cloud-sliced retrievals exceed 200 pptv. Seasonal means are obtained by Gaussian weighting individual estimates of cloud-sliced UT $NO_2$ to the pressure centre (315 hPa).

The cloud-slicing retrieval adopted here is mostly similar to that applied to OMI to estimate mid-tropospheric $NO_2$ at 900-650

hPa (Choi et al., 2014) and UT $NO_2$ at 450-280 hPa (Marais et al., 2018). We extend the ceiling of the retrieval to 180 hPa (~12.5 km) to better capture the vertical extent of the upper troposphere. Another notable distinction is that the method applied to OMI used vertical gradients of $NO_2$ from the NASA Global Modeling Initiative (GMI) CTM to diagnose scenes with non-uniform $NO_2$ using a threshold of 0.33 pptv $hPa^{-1}$. We dispense with this step, as its application to TROPOMI requires a model at a similar fine spatial resolution to TROPOMI and CTMs may underestimate vertical $NO_2$ gradients in the UT (Boersma et

al., 2011; Travis et al., 2016; Silvern et al., 2018). Anyway, we find that the strict filtering applied to GEOS-Chem partial columns removes most (88%) scenes with $NO_2$ vertical gradients $\geq 0.33$ pptv $hPa^{-1}$.

Figure 1 shows GEOS-Chem seasonal mean cloud-sliced and "true" cloudy UT $NO_2$ at $4° \times 5°$. The latter is also Gaussian weighted to 315 hPa. The uncertainty in individual cloud-sliced values, estimated as the RMA regression slope error, range

from 6% to the imposed error limit, 99%. This is reduced to <2% for the multiyear seasonal means in Figure 1 due to temporal averaging. Agreement between the cloud-sliced and "true" cloudy UT $NO_2$ is shown in the scatterplot in Figure 2. Successful cloud-sliced retrievals can exceed 35 for many grid squares, though these do not exhibit better agreement with the "truth" than



the grid squares with fewer (<10) retrievals. The two datasets are spatially consistent (R = 0.64) and exhibit similar variance (slope = 1.1 ± 0.1). The cloud-sliced UT $NO_2$ has a small positive offset in background UT $NO_2$ (intercept = 2.3 ± 1.2 pptv).

On average, cloud-sliced UT $NO_2$ is 17% more than the "true" cloudy UT $NO_2$, but this depends on the spatial resolution of the retrieved cloud-sliced product. Regression slopes increase from 0.87 ± 0.03 for cloud-sliced UT $NO_2$ obtained at 2° × 2.5° to 1.4 ± 0.2 at 8° × 10° and the cloud-sliced UT $NO_2$ is 4.1% less than the "true" cloudy UT $NO_2$ at 2° × 2.5° and 37% more at 8° × 10°. Maps of synthetic cloud-sliced UT $NO_2$ at 2° × 2.5° and 8° × 10° are in Figure S1. Strict data filtering in the cloud-slicing steps removes 90% of the clusters of GEOS-Chem partial columns for the 4° × 5° product. Most (33%) data loss is due

to the strict relative standard deviation threshold applied to stratospheric $NO_2$. Cloud-slicing is very sensitive to this threshold. Relaxing it from a relative standard deviation of 0.02 to 0.03 increases data retention from 10% of the clusters of GEOS-Chem partial columns to 17%, but increases the positive bias in cloud-sliced UT $NO_2$ from 17% to 45%, This is due to an increase in the contribution of variability in the stratosphere to the cloud-slicing regression slopes.

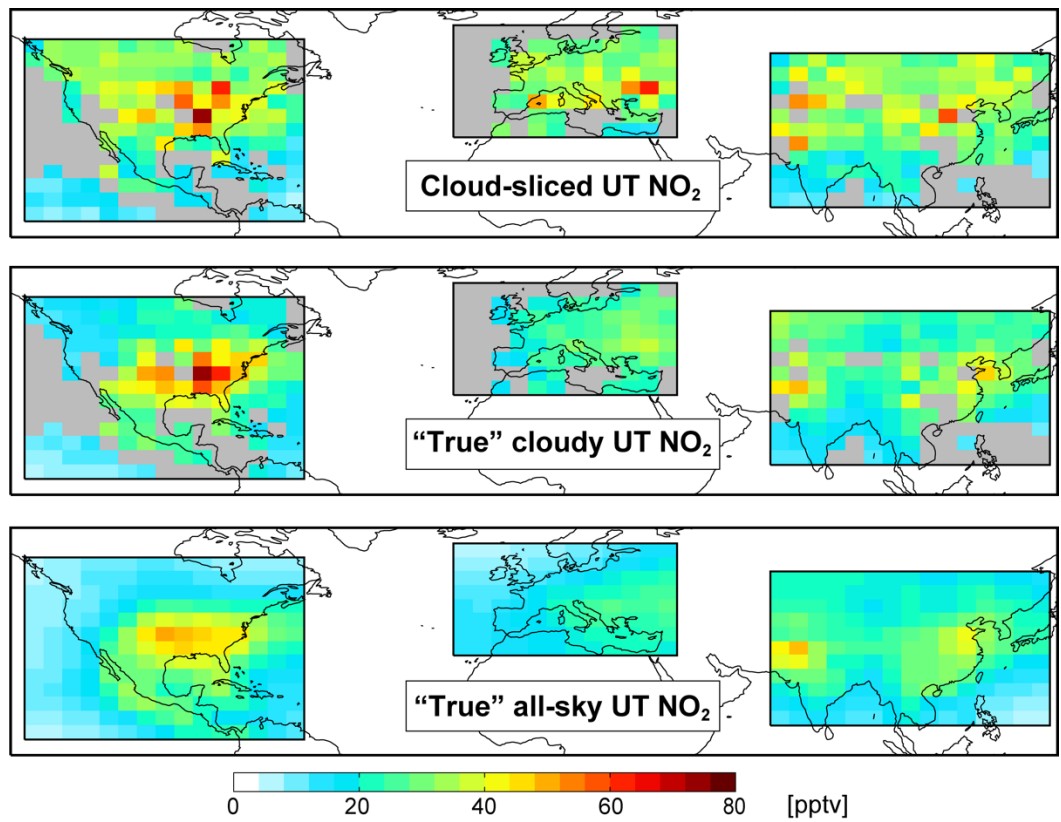


**Figure 1: Comparison of synthetic cloud-sliced and "true" $NO_2$ in the upper troposphere (UT) for June-August 2016-2017. Maps show UT $NO_2$ at 4° × 5° from cloud-slicing GEOS-Chem partial columns above all clouds with cloud top pressures at 450-180 hPa (top), as grid-average mixing ratios from GEOS-Chem for the same scenes as are cloud-sliced (middle) and for all-sky (clear and cloudy) scenes (bottom). Data are Gaussian weighted to the pressure centre (315 hPa). Grey grids have <5 data points.**





Also shown in Figure 1 is the "true" all-sky UT $NO_2$ obtained for all (cloudy and clear) scenes across 450-180 hPa. Model grids with stratospheric influence are identified and removed using GEOS-FP tropopause heights that are updated hourly in the model. The "true" cloudy UT $NO_2$ is 17% more than all-sky UT $NO_2$. Spatial resolution influences the size of this difference, increasing from 11% at 2° × 2.5° to 22% at 8° × 10°. This suggests that isolating cloudy scenes induces a 11-22% bias in seasonal mean $NO_2$ that could be due to a combination of poor data retention (low sampling frequency of cloudy

scenes), the influence of clouds on $NO_x$ photochemistry (Pour-Biazar et al., 2007; Holmes et al., 2019), and local enhancements in $NO_x$ from events like lightning and deep convective uplift of surface pollution that accompany clouds (Crawford et al., 2000; Ridley et al., 2004; Bertram et al., 2007).

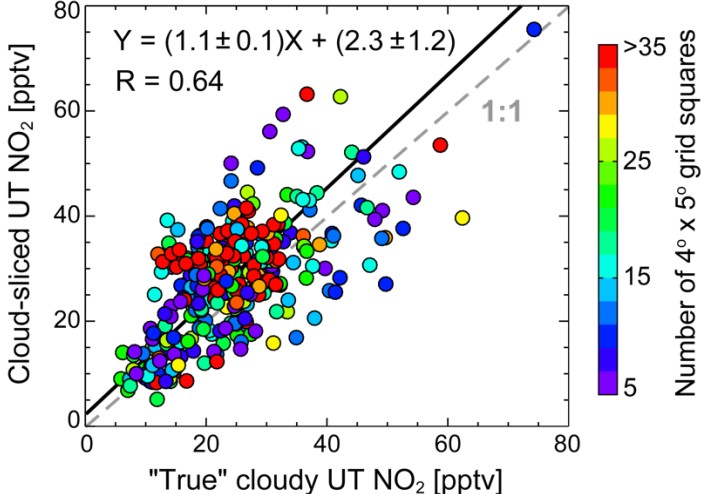

**Figure 2: Scatter plot of synthetic cloud-sliced versus "true" cloudy $NO_2$ in the upper troposphere (UT). Points are 4° × 5° seasonal means from Figure 1 (top and middle panels) coloured by the number of successful cloud-sliced retrievals. Values inset are the RMA regression statistics and Pearson's correlation coefficient (R). Slope and intercept errors are from bootstrap resampling.**

    Cloud-slicing applied to GEOS-Chem considers all cloudy scenes, whereas cloud-slicing of satellite observations is applied to partial columns above optically thick clouds to minimise contamination of $NO_2$ from below the cloud. If we only consider

synthetic partial columns above clouds with a physical (geometric) cloud fraction across 450-180 hPa of at least 0.7, the cloud-sliced UT $NO_2$ positive difference is similar (18%) to that obtained for all cloudy scenes, but half the amount of data is retained. The cloud fraction retrieved from TROPOMI is an effective or radiometric cloud fraction that is systematically less than the physical cloud fraction from the model. Our results suggest that representation error is not sensitive to the cloud fraction threshold. Another distinction in GEOS-Chem and TROPOMI cloud variables is that the model provides the physical cloud

top height, whereas TROPOMI cloud retrievals that use models that assume clouds are uniform reflective boundaries retrieve cloud top heights that can be ~1 km lower than the physical cloud top (Joiner et al., 2012; Choi et al., 2014; Loyola et al., 2018a). We again apply the cloud-slicing algorithm to the simulated partial columns, but with the cloud top heights artificially reduced by 1 km. This approach assumes that the difference in altitudes of effective (radiometric) clouds and physical clouds



is systematic and vertically and horizontally uniform. The difference between the resultant cloud-sliced UT $NO_2$ and the "true"

cloudy UT $NO_2$ shown in the middle panel of Figure 1 increases from 17% to 24%. This is because the decrease in cloud top altitude leads to a larger increase in the vertical extent of partial columns above high-altitude clouds than those above low-altitude clouds leading to steeper regression slopes and larger UT $NO_2$.

## 3 Evaluation of TROPOMI with ground-based instruments at high-altitude sites

Pandora spectrometer systems provide observations of total and tropospheric columns of $NO_2$ using direct sun, direct moon

and sky radiance observations (Herman et al., 2009; Cede et al., 2019). Those at high-altitude sites have limited influence from the planetary boundary layer and so are used here to evaluate free tropospheric and stratospheric $NO_2$ from TROPOMI. These include long-term Pandora instruments at Mauna Loa, Hawaii (19.48°N, 155.60°W, 4.2 km above sea level or a.s.l, ~600 hPa), Izaña, Tenerife, Canary Islands (28.31°N, 16.50°W, 2.4 km a.s.l, ~760 hPa), and Altzomoni, Mexico (19.12°N, 98.66°W, 4.0 km a.s.l, ~620 hPa). Mauna Loa and Izaña are remote and have limited anthropogenic influence (Toledano et al., 2018),

whereas Altzomoni is ~70 km southeast of Mexico City and is often within the mixed layer of the city in the afternoon (Baumgardner et al., 2009) after the TROPOMI overpass. On average, multiyear mean tropospheric $NO_2$ columns from OMI are ~$10 \times 10^{15}$ molecules $cm^{-2}$ lower over Altzomoni (<$5 \times 10^{15}$ molecules $cm^{-2}$) than the city (>$15 \times 10^{15}$ molecules $cm^{-2}$) (Rivera et al., 2013). At Izaña, there is also a MAX-DOAS instrument that we use to retrieve tropospheric columns of $NO_2$ to assess Pandora and TROPOMI. MAX-DOAS offers vertical sensitivity in the troposphere and has been used extensively to

determine free tropospheric concentrations of $NO_2$ at high-altitude sites (Gomez et al., 2014; Gil-Ojeda et al., 2015; Schreier et al., 2016).

Pandora level 2 total and tropospheric columns are from the Pandonia Global Network (PGN) ([http://data.pandonia-global-network.org/](http://data.pandonia-global-network.org/); last accessed 1 June 2020). We use version 1.7 "nvh1" retrieval of total columns and "nvs1" retrieval of

tropospheric columns (described below). Observations are for a full year (1 June 2019 to 31 May 2020) at Izaña. The data record is shorter at Mauna Loa (ends 29 March 2020) and Altzomoni (ends 9 March 2020). Total slant columns ($NO_2$ abundances along the instrument viewing path) are retrieved by fitting a fourth order polynomial to spectra at 400-440 nm using an $NO_2$ effective temperature of 254.4 K. These are then converted to total vertical column densities by accounting for the geometry of the viewing path (Cede et al., 2019). The Pandora tropospheric $NO_2$ columns have not yet been validated

against other observations. Retrieval of these involves simultaneous retrieval of slant columns of $NO_2$ and the $O_2$-$O_2$ dimer at multiple elevation angles (typically 0°, 60°, 75°, 88°, and 89°). The $O_2$-$O_2$ dimer slant columns are used to calculate a representative air mass factor (AMF) that is applied to the difference in $NO_2$ slant columns at multiple pointing elevation angles to calculate a tropospheric vertical column. The data also include estimates of the uncertainty on the total and tropospheric columns due to instrument noise and atmospheric variability (Cede et al., 2019). The $NO_2$ effective temperature

used in the total $NO_2$ column retrieval is greater than the column average ambient temperature at high-altitude sites. This



induces a positive bias in the total columns estimated by Verhoelst et al. (2020) to be ~10% that we address by downscaling the Pandora total columns and associated errors by 10%. No correction is applied to the tropospheric columns, due to variable contribution of the troposphere to the total column.

MAX-DOAS vertical tropospheric columns of $NO_2$ at Izaña are from RASAS-II sky radiance spectra for June 2019 to February 2020. The spectra are fitted for $NO_2$ and $O_2$-$O_2$ in the wavelength range 425-490 nm and slant columns are calculated as the difference between these spectra at high-sun (90° instrument elevation angle) and multiple elevation angles (1°, 2°, 3°, 5°, 10°, 30°, and 70°) (Hönninger et al., 2004; Gil et al., 2008; Puentedura et al., 2012; Gomez et al., 2014; Gil-Ojeda et al., 2015). Vertical columns are estimated using optimal estimation that solves an ill-constrained problem by introducing prior

information (Rogers, 2000). Prior information for Izaña includes fixed (with altitude) aerosol extinction of 0.01 $km^{-1}$ and $NO_2$ of 20 pptv from the surface to the tropopause. Aerosol abundances at Izaña are sometimes influenced by windblown dust from the Sahara Desert, but are typically low (aerosol optical depth or AOD < 0.05) (Gomez et al., 2014; Gil-Ojeda et al., 2015). The prior $NO_2$ profile is within the range of background $NO_2$ in the UT (10-20 pptv) (Marais et al., 2018) and MAX-DOAS $NO_2$ concentrations previously retrieved at Izaña (20-40 pptv) (Gomez et al., 2014). Filtering is applied to remove vertical

column retrievals with limited independent information (degrees of freedom for signal < 1), and significant light path attenuation by aerosols (AOD > 0.1) and clouds (effective cloud fraction > 0.5). AOD is derived with MAX-DOAS $O_2$-$O_2$ dimer differential slant columns retrieved over the same wavelength range as $NO_2$ (Frieß et al., 2006) and cloud fraction is from the Fast Retrieval Scheme for Clouds from the Oxygen A band version S (FRESCO-S) product provided with the TROPOMI $NO_2$ product. Filtering removes 40% of the retrieved vertical tropospheric $NO_2$ columns at Izaña.


TROPOMI data is from the Sentinel-5P Pre-Operations Data Hub (https://s5phub.copernicus.eu/dhus/; last accessed 15 June 2020). We use a year of $NO_2$ data (1 June 2019 to 31 May 2020) from the level 2 offline (OFFL) product version 01-03-02. The data product includes $NO_2$ abundances along the optical path from the sun to the instrument (the total slant column or $SCD_{tot}$), $NO_2$ vertical column densities in the stratosphere ($VCD_{strat}$), and the stratospheric air mass factor ($AMF_{strat}$). A detailed

description of retrieval of $SCD_{tot}$ and $VCD_{strat}$ is described in the product Algorithm Theoretical Basis Document (van Geffen et al., 2019) and by van Geffen et al. (2020). In brief, $SCD_{tot}$ are obtained by spectral fitting of TROPOMI top-of-atmosphere radiances at 405-465 nm by accounting for light absorption by $NO_2$ and other relevant gases. $VCD_{strat}$ are from assimilation of TROPOMI and modelled total slant columns over locations diagnosed by the model to have limited tropospheric influence (predominantly remote oceans) (Boersma et al., 2004; Dirksen et al., 2011; van Geffen et al., 2019). The modelled slant

columns are the product of vertical columns from the TM5-MP CTM (Williams et al., 2017) and AMFs calculated using TROPOMI viewing geometries and surface reflectivities. The CTM is simulated at 1° × 1° and driven with ECMWF meteorology updated every 3 hours. $SCD_{tot}$ are separated into a stratospheric ($SCD_{strat}$) and tropospheric ($SCD_{trop}$) component and a tropospheric AMF ($AMF_{trop}$) is applied to $SCD_{trop}$ to obtain tropospheric vertical columns ($VCD_{trop}$). $AMF_{trop}$ accounts for viewing geometry, surface reflectivity, atmospheric absorption and scattering of light by trace gases and aerosols, and





sensitivity to the vertical distribution of $NO_2$. A vertically resolved correction is also applied to the $AMF_{trop}$ to correct for the fixed $NO_2$ effective temperature (220 K) used to retrieve $SCD_{tot}$. The light path in the UT is relatively unobstructed by aerosols and, for cloud-slicing, would mostly be impacted by treatment of the reflectivity of optically thick clouds. We choose to use an AMF that only accounts for viewing geometry ($AMF_{trop,geo}$) due to uncertainties in the modelled vertical distribution of $NO_2$ in the UT (Stavrakou et al., 2013; Travis et al., 2016) and representation errors from a model at coarser resolution (~100 km)

than TROPOMI (< 10 km at nadir). Choi et al. (2014) found that OMI partial $NO_2$ columns calculated with $AMF_{trop,geo}$ above optically thick clouds in the mid-troposphere (650 hPa) were at most 14% more than those calculated with a detailed AMF that assumed clouds are near-Lambertian surfaces with albedo of 0.8 and $NO_2$ is constant with altitude. The effect of not including a temperature correction will be small in the UT where temperatures are ~220 K anyway. To confirm this, we find that GEOS-Chem cloud-sliced UT $NO_2$ calculated with the TROPOMI AMF temperature correction expression in van Geffen

et al. (2019) are only 6% less than those in Figures 1-2.

We calculate $VCD_{trop}$ by first obtaining $SCD_{trop}$ as the difference between $SCD_{tot}$ from the data product and $SCD_{strat}$ calculated as the product of the reported $VCD_{strat}$ and $AMF_{strat}$:

$$SCD_{trop} = SCD_{tot} - (VCD_{strat} \times AMF_{strat}) \qquad (1).$$

This we use to estimate the above-cloud $VCD_{trop}$ using $AMF_{trop,geo}$ calculated with the reported solar zenith angles (SZA) and viewing zenith angles (VZA):

$$VCD_{trop} = \frac{SCD_{trop}}{AMF_{trop,geo}} = \frac{SCD_{trop}}{(\sec(SZA)+\sec(VZA))} \qquad (2).$$

The TROPOMI $VCD_{tot}$ we compare to Pandora are calculated as the sum of reported $VCD_{strat}$ and our calculated $VCD_{trop}$ (Equation (2)). Only data with quality flags ("qa_value" in the data product) of at least 0.45 are used. This removes data

affected by sun glint, poor precision in the retrieval and radiances, and SZA > 84.5° (van Geffen et al., 2019). Similarly, good quality Pandora retrievals of total and tropospheric columns are identified as those with data quality flags of 0, 1, 10, or 11 (Cede et al., 2019), consistent with Ialongo et al. (2020). Coincident satellite and ground-based data are identified as TROPOMI pixels within a 0.2° radius (~20 km) of the station and ground-based data ±30 min around the TROPOMI overpass.

The upper panel of Figure 3 compares collocated daily mean Pandora and TROPOMI total columns. Errors on the daily means, obtained by adding in quadrature reported uncertainties of individual columns, are small at all sites. These vary from 0.1% to 19% for Pandora and 1.5% to 16% for TROPOMI. TROPOMI and Pandora total columns are temporally consistent (R = 0.69 at Mauna Loa, R = 0.87 at Izaña, R = 0.67 at Altzomoni), but there is a systematic positive offset in TROPOMI ranging from $6.6 \times 10^{14}$ molecules cm$^{-2}$ at Mauna Loa to $9.3 \times 10^{14}$ molecules cm$^{-2}$ at Altzomoni and TROPOMI is on average 18% higher

than Pandora at Mauna Loa, 26% at Izaña, and 38% at Altzomoni. Verhoelst et al. (2020) also report a positive bias in TROPOMI total columns at the same Pandora sites of 6% at Mauna Loa, 19% at Izaña, and 28% at Altzomoni for April 2018





to February 2020. Our higher values compared to Verhoelst et al. (2020) is because of the 10% downscaling we apply to Pandora total columns. The difference in sampling footprints of space- and ground-based instruments can influence agreement between the two (Pinardi et al., 2020). We find though that the difference between TROPOMI and Pandora at Mauna Loa and

Izaña is relatively unchanged by the choice of sampling coincidence. The difference is 17-20% at Mauna Loa and 25-26% at Izaña for a TROPOMI sampling radius of 0.05-0.3° and for a Pandora sampling time window of ±15-60 min. The comparison at Altzomoni though is very sensitive to the sampling radius due to proximity to Mexico City. There the difference increases from 22% at 0.05° for 45 coincident points to 48% at 0.3° for 76 coincident points.

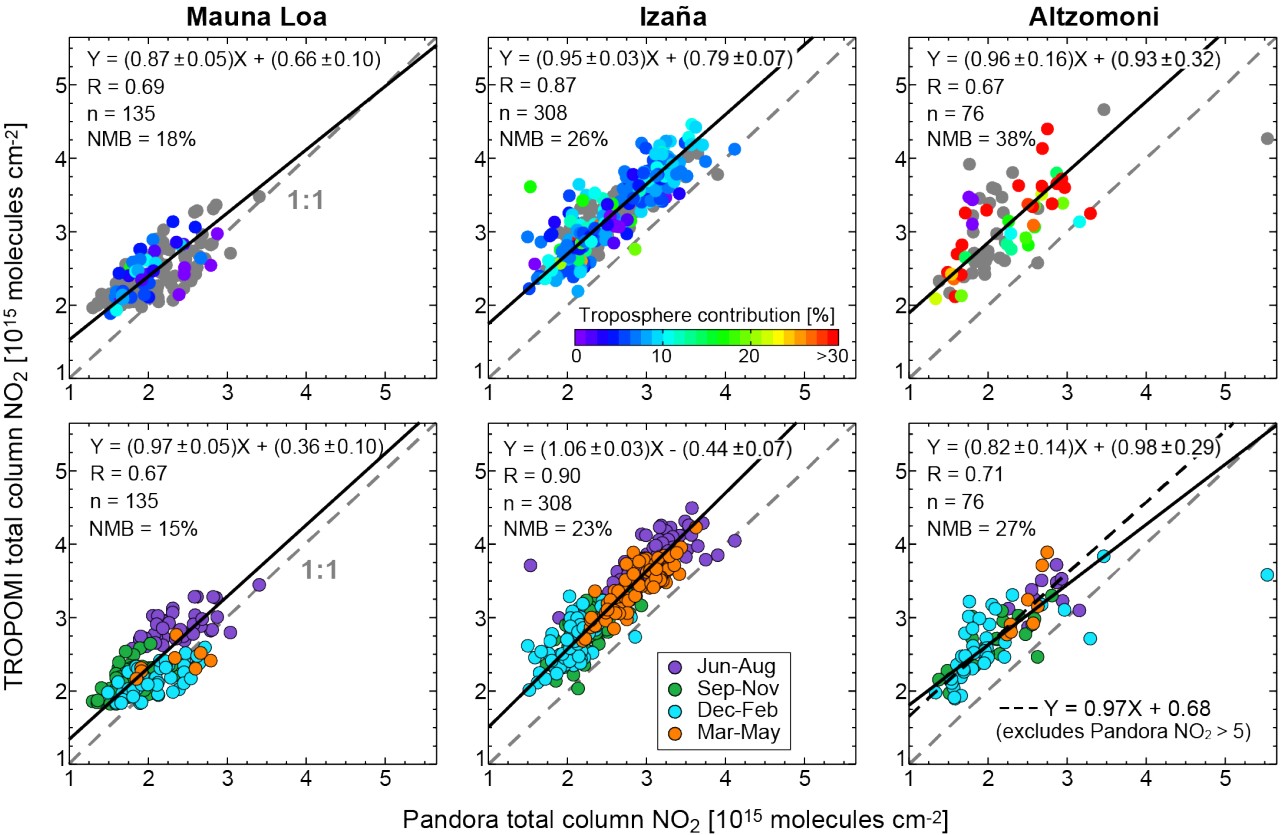


**Figure 3: Comparison of TROPOMI and Pandora total NO₂ columns at high-altitude sites. Points are daily means with at least 5 coincident observations at Mauna Loa (left), Izaña (centre), and Altzomoni (right) before (upper) and after (lower) applying correction factors to TROPOMI stratospheric and tropospheric columns (see text for details). Upper panel colours are the relative contribution of the troposphere to the total column according to Pandora where available, grey otherwise. Data in the lower panel**
**are coloured by season. Lines are the 1:1 relationship (grey dashed) and RMA regression (black solid). Values inset are Pearson's correlation coefficients, RMA regression statistics, number of data points (n), and the TROPOMI normalized mean bias (NMB). Also shown for Altzomoni (bottom right panel) is the RMA regression without the Pandora > 5 × 10¹⁵ molecules cm⁻² (black dashed line). Axes do not start at the origin.**

At Mauna Loa, the tropospheric column contribution to the total averages 5.1% (range of 0.2-16%), according to Pandora,
compared to 8.3% (0.2-38%) at Izaña and 31% (8-91%) at Altzomoni. We thus use Mauna Loa total columns to identify that



TROPOMI underestimates stratospheric $NO_2$ variance by 13% (slope = 0.87 ± 0.05). This is likely because the variability in stratospheric $NO_2$ is smoothed by the coarser spatial resolution of the TM5-TMP model (1° × 1°) and time resolution of the meteorology (3-hourly). The underestimate in stratospheric $NO_2$ variance would lead to an overestimate in the relative contribution of the stratosphere to the total column for small column densities and vice versa. The impact on the cloud-sliced

UT $NO_2$ is steep regression slopes and an overestimate in cloud-sliced UT $NO_2$, as the upper troposphere column density will be overestimated for high-altitude clouds and underestimated for low-altitude clouds. The 18% higher TROPOMI than Pandora total columns at Mauna Loa is larger than and opposite in sign to the <10% (-2 × $10^{14}$ molecules $cm^{-2}$) meridional difference in TROPOMI stratospheric columns from the near-real time (NRTI) $NO_2$ product and those obtained with twilight measurements from the near-global Système d'Analyse par Observation Zénitale (SAOZ) network of Zenith Scattered Light

Differential Optical Absorption Spectroscopy (ZSL-DOAS) instruments (Lambert et al., 2019). The implied difference between SAOZ and Pandora stratospheric columns coincident with TROPOMI (Pandora < SAOZ) may be due to the need to account for time differences between the SAOZ measurements (twilight) and TROPOMI (midday) (Verhoelst et al., 2020). This difference warrants further investigation, as these ground-based measurements are crucial for validating space-based sensors that measure $NO_2$.


The underestimate in TROPOMI stratospheric column variance may contribute to the general pattern in validation studies comparing TROPOMI and Pandora total columns that find TROPOMI is less than Pandora when $NO_2$ is large and more than Pandora when $NO_2$ is small for the global Pandora network (Pinardi et al., 2020; Verhoelst et al., 2020) and at individual Pandora sites. TROPOMI is less than Pandora (-24 to -28%) at a relatively polluted Greater Toronto Area site, but more than

Pandora (8-11%) at a cleaner rural site north of the city (X. Zhao et al., 2020). Similarly, at a site in Helsinki, Finland, TROPOMI is less than Pandora (-28%) for Pandora > 10 × $10^{15}$ molecules $cm^{-2}$ and more than Pandora (17%) for Pandora < 10 × $10^{15}$ molecules $cm^{-2}$ (Ialongo et al., 2020).

After applying the stratospheric $NO_2$ variance correction, the intercepts in the top panel of Figure 3 decrease to 4.4 × $10^{14}$

molecules $cm^{-2}$ for Mauna Loa, 7.9 × $10^{14}$ molecules $cm^{-2}$ for Izaña, and 7.3 × $10^{14}$ molecules $cm^{-2}$ for Altzomoni (not shown). Likely causes for the remaining discrepancy between TROPOMI and Pandora include a positive offset in the TROPOMI radiance intensity that is 5% of the total column or 0.1-1 × $10^{15}$ molecules $cm^{-2}$ (van Geffen et al., 2020), challenges obtaining a Pandora reference measurement (atmospheric column without $NO_2$) (Herman et al., 2009), and an overestimate in TROPOMI free tropospheric $NO_2$. The radiance intensity offset has been shown to mostly affect retrievals over open oceans (van Geffen

et al., 2020), and an overestimate in free tropospheric $NO_2$ would have a larger effect on the total column comparison at Izaña and Altzomoni than at Mauna Loa.





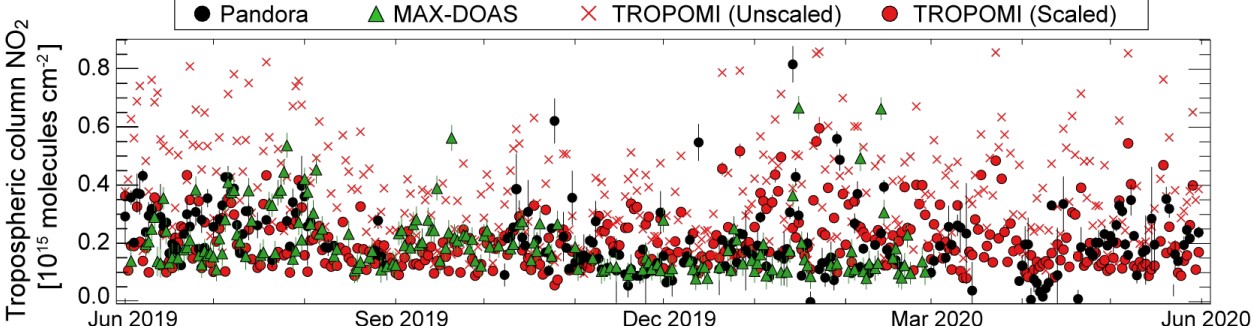

**Figure 4: Time series of free tropospheric column NO₂ at Izaña.** Points are daily midday means from Pandora (black circles), MAX-DOAS (green triangles), and TROPOMI (red) before (crosses) and after (circles) applying scaling factors to the stratospheric and tropospheric columns (see text for details). Error bars are individual retrieval uncertainties added in quadrature.

Figure 4 compares time series of free tropospheric NO₂ at Izaña from Pandora, MAX-DOAS and TROPOMI. As with the total columns, Pandora and MAX-DOAS are sampled 30 min around the satellite overpass and TROPOMI 0.2° around the site. We impose a modest threshold to sample TROPOMI tropospheric columns $>4 \times 10^{13}$ molecules cm$^{-2}$ to mimic the detection limits of the instruments (Gomez et al., 2014) and mitigate the influence of TROPOMI data that would be susceptible to errors in distinguishing the stratosphere from the troposphere. This brings the lower-end TROPOMI values into better agreement with the ground-based values and has no effect on TROPOMI columns $>2 \times 10^{14}$ molecules cm$^{-2}$. On average, Pandora is 14% more than MAX-DOAS and the temporal correlation is modest (R = 0.4). Temporal inconsistencies between Pandora and MAX-DOAS are due to challenges retrieving tropospheric columns routinely close to instrument detection limits (Gomez et al., 2014), lack of dynamic variability in the retrieved columns, and differences in the sampling extent of the two instruments. The MAX-DOAS sampling footprint, for example, shifts by at least 2° in latitude between winter and summer solstices (Robles-Gonzalez et al., 2016). Most MAX-DOAS and Pandora data are at $1-4 \times 10^{14}$ molecules cm$^{-2}$, whereas the range for TROPOMI calculated using Eq. (1) and (2) extends to $\sim 8 \times 10^{14}$ molecules cm$^{-2}$. P. Wang et al. (2020a) obtained the same range in tropospheric column densities from comparison of TROPOMI to shipborne MAX-DOAS measurements. They found that TROPOMI was on average $4 \times 10^{14}$ molecules cm$^{-2}$ more than MAX-DOAS. In our comparison, TROPOMI free tropospheric columns (red crosses in Figure 4) are 77% more than Pandora and 84% more than MAX-DOAS. A similar overestimate is obtained if the reported detailed tropospheric AMF is used instead of AMF$_{trop,geo}$ (Eq. (2)) to calculate TROPOMI tropospheric columns. The stratospheric variance correction reduces the difference between TROPOMI and the ground-based measurements to 40% compared to Pandora and 47% compared to MAX-DOAS due to an increase in the relative contribution of the stratosphere to total columns $> 2 \times 10^{15}$ molecules cm$^{-2}$. To address the remaining difference between TROPOMI tropospheric columns and the ground-based observations, we downscale TROPOMI tropospheric columns by 50% (red circles in Figure 4) leading to a difference of -4% with Pandora and 1% with MAX-DOAS. There is no temporal correlation between daily coincident observations of TROPOMI and the ground-based measurements (R < 0.1), consistent with the comparison of TROPOMI to shipborne MAX-DOAS by P. Wang et al. (2020a).



385

The lower panel in Figure 3 compares Pandora to TROPOMI total columns after increasing TROPOMI stratospheric column variance by 13% and reducing TROPOMI tropospheric columns by 50%. This correction reduces the difference between TROPOMI and Pandora by just 3 percentage points at Mauna Loa and Izaña and 11 percentage points at Altzomoni. The variance at Altzomoni degrades from $0.96 \pm 0.16$ to $0.82 \pm 0.14$, but this is because the relatively few coincident points (76 compared to 308 at Izaña) are influenced by the single Pandora observation $>5.5 \times 10^{15}$ molecules cm$^{-2}$ (coincident corrected TROPOMI is $< 4 \times 10^{15}$ molecules cm$^{-2}$) that may be detecting $NO_2$ from fires typical of December-February in the National Park where the instrument is located (Bravo et al., 2002; Baumgardner et al., 2009). The TROPOMI tropospheric column contribution at Mauna Loa and Izaña is more consistent with that from Pandora after applying the stratospheric and tropospheric column corrections, decreasing from 8% to 6% at Mauna Loa and 12% to 7% at Izaña. This is not the case for Altzomoni (decrease from 14% to 9%), due to anthropogenic influence from Mexico City. Points in Figure 3 are coloured by season to show that all sites experience a modest decline in $NO_2$ from summer (purple) to winter (cyan) due to the influence of solar variability on photochemical production of $NO_x$ in the stratosphere (Gil et al., 2008; Robles-Gonzalez et al., 2016) and seasonality in long-range transport and subsidence in the free troposphere (Gil-Ojeda et al., 2015). The distinct distribution of points in December-February compared to June-August and September-November at Mauna Loa suggest there may be seasonality in the size of the discrepancy between TROPOMI and Pandora stratospheric columns. The remaining TROPOMI positive offset of $\sim 4 \times 10^{14}$ molecules cm$^{-2}$ is consistent with the $2\text{-}4 \times 10^{14}$ molecules cm$^{-2}$ positive offset in TROPOMI stratospheric columns reported by P. Wang et al. (2020a) from comparison to shipborne MAX-DOAS measurements. If the remaining offset is exclusively due to the stratospheric column, this would cancel in the cloud-slicing retrieval for clusters of partial columns with uniform stratospheric $NO_2$.

## 4 Retrieval of TROPOMI NO₂ in the upper troposphere

The same cloud-slicing retrieval steps applied to synthetic spectra from GEOS-Chem (Section 2) are applied to corrected TROPOMI total columns to obtain seasonal mean UT $NO_2$ for a year (June 2019 to May 2020) at $1° \times 1°$. This resolution degrades TROPOMI nadir pixels by 400-fold compared to 250-fold for the synthetic experiment in Section 2 and a much greater (1300-fold) degradation in OMI nadir pixel resolution (13 km $\times$ 24 km) for the $5° \times 8°$ product (Marais et al., 2018). The finer relative resolution we choose for TROPOMI cloud-sliced UT $NO_2$ compared to OMI is informed by the synthetic experiment applied to GEOS-Chem and the superior cloud-resolving capability of TROPOMI than OMI. Cloud-slicing is applied to partial columns above optically thick clouds (diagnosed with an effective cloud fraction $\geq 0.7$, as in Marais et al. (2018)) to limit contamination from light transmitted through clouds. Though the cloud-slicing retrieval steps applied to GEOS-Chem and TROPOMI are the same, there are differences in the modelled and retrieved cloud parameters that we discuss below.



Two TROPOMI cloud-sliced UT $NO_2$ products are derived using cloud top heights and cloud fractions from distinct cloud products. These are FRESCO-S from the same data file as TROPOMI $NO_2$ and the standalone offline (OFFL) cloud product. FRESCO-S cloud fractions and cloud top pressures are determined by minimizing the difference between measured and simulated spectra in the $O_2$ A-band (758-766 nm) using lookup tables of relevant physical parameters and assuming clouds are single layer Lambertian reflectors with albedo of 0.8 (P. Wang et al., 2008; van Geffen et al., 2019). The standalone product version number changes from 01-01-07 to 01-01-08 on 7 March 2020, but with no change to the data product (Argyrouli et al., 2019). Cloud fractions are from the Optical Cloud Recognition Algorithm (OCRA). In OCRA, cloud fractions are retrieved by determining the difference, in colour space, between cloudy and clear reflectances in blue (405-495 nm) and green (350-395 nm) broad spectral bands (Loyola et al., 2007; 2018a; 2018b). Cloud top heights (in km) are from the Retrieval of Cloud Information using Neural Networks (ROCINN). This involves minimizing the difference between measured $O_2$ A-band radiances and neural network trained radiances modelled using OCRA cloud fractions as input (Loyola et al., 2007; 2018b) and assuming clouds are multiple optically uniform layers of light-scattering water droplets (the clouds-as-layers or CAL model). We convert ROCINN-CAL cloud top heights to pressures for cloud-slicing and comparison to FRESCO-S. FRESCO-S data are quality screened using the same qa_value threshold (0.45) as the $NO_2$ data. A qa_value threshold of 0.5 is used for OCRA cloud fractions and ROCINN-CAL cloud heights. This removes data affected by sun glint, spectral saturation that is particularly problematic over bright high-altitude clouds and not properly flagged in the level 1 radiances used to retrieve $NO_2$ slant columns (Gorkavyi et al., 2020), poor quality radiances and retrievals, SZA > 75°, and issues arising from spatial misalignment of ground pixels of different spectral bands (Loyola et al., 2018a; 2018b). Snow/ice scenes that could be misidentified as clouds in the FRESCO-S product are identified as scenes with differences in reported scene and surface pressures > 2% (as in van der A et al. (2020)), snow cover > 80% or permanent ice cover. Snow and ice cover and classification are from the Near-real-time Ice and Snow Extent (NISE) product provided with the $NO_2$ product (van Geffen et al., 2019). For OCRA and ROCINN-CAL, we use the reported snow/ice flag that combines information from NISE and a surface albedo climatology (Loyola et al., 2018b). In what follows, we distinguish the two cloud-sliced TROPOMI UT $NO_2$ products as FRESCO-S UT $NO_2$ and ROCINN-CAL UT $NO_2$.



**Figure 5: Seasonal mean upper tropospheric (UT) NO₂ from TROPOMI. Maps are UT NO$_2$ at 1° × 1° in June-August 2019 (first row), September-November 2019 (second), December 2019 to February 2020 (third), and March-May 2020 (fourth) using FRESCO-S (left) or ROCINN-CAL (right) cloud information and with corrections applied to TROPOMI stratospheric and tropospheric columns (see text for details). Inset numbers give total successful cloud-sliced retrievals. Grey grid squares have fewer than 5 cloud-sliced retrievals.**

Figure 5 shows maps of seasonal mean FRESCO-S and ROCINN-CAL UT NO$_2$ at 1° × 1°. The spatial features are consistent with a combination of the density of lightning flashes (Cecil et al., 2014) and lightning properties such as flash footprint, duration, and energy (Beirle et al., 2014). These include elevated concentrations (> 80 pptv) over northern hemisphere land masses in June-August, the year-round 40-60 pptv band over tropical landmasses that shifts meridionally with the Intertropical Convergence Zone (ITCZ), and relatively low concentrations (< 30 pptv) over the remote Pacific Ocean. In the cold polar regions UT NO$_2$, limited to ROCINN-CAL, are near-background (<30 pptv; Marais et al. (2018)) as NO$_2$ is preferentially present as NO$_x$ reservoir compounds such as peroxyacetyl nitrates (PANs) (Bottenheim et al., 1986). Large enhancements (NO$_2$ > 80 pptv) over northern China and the northeast US in June-August, and Australia in December-February most prevalent in the ROCINN-CAL product likely reflect contamination from surface pollution below clouds. These would result from





intense anthropogenic activity in North China and the northeast US (B. Zhao et al., 2013; Jiang et al., 2018; Z. Wang et al., 2020b) and routine pyrocumulonimbus injection of fire plumes into the free troposphere and lower stratosphere during the intense 2019-2020 fire season in Australia (Kablick III et al., 2020). Coincident gridsquares of the two data sets are spatially

consistent (R of 0.82 to 0.88), though ROCINN-CAL UT $NO_2$ are 4.2-9.1 pptv more than FRESCO-S UT $NO_2$. As with the synthetic experiment, UT $NO_2$ increases with degradation in resolution. Depending on the season, cloud-sliced UT $NO_2$ are 2-4% more at $2° \times 2.5°$ and 3-9% more at $4° \times 5°$ than at $1° \times 1°$. Good quality retrievals and optically thick clouds with cloud top pressures at 450-180 hPa account for ~2% of TROPOMI pixels using FRESCO-S and ~3% using ROCINN-CAL. Of these, 44,000-78,000 cloud-slicing retrievals are retained in each season for FRESCO-S, and 118,000-177,000 for ROCINN-CAL

(Figure 5). Most data loss in the cloud-slicing retrieval is because of too few points (clusters < 10) or cloud top pressure range < 140 hPa. Discarded extreme values of cloud-sliced $NO_2$ > 200 pptv are only 0.1-0.5% of retained data. More cloud-sliced retrievals with ROCCIN-CAL is due to greater abundance of optically thick clouds and clusters with greater cloud height range.

Figure 6 compares the meridional abundance of optically thick clouds in the UT from the two cloud products for June-August and December-February. The same information for the other two seasons is in Figure S2. Both products retrieve effective (radiometric) cloud fractions. These are systematically less than the physical (geometric) cloud fractions from GEOS-Chem, though the two converge for optically thick clouds with physical cloud fractions approaching 1 (Stammes et al., 2008). The number of OCRA optically thick clouds is always more (often double) than that of FRESCO-S in all seasons and across all

latitudes. The greatest difference in the number of optically thick clouds tracks the ITCZ and is also typically at 45°N/S. The majority (61-62%) of OCRA cloud fractions exceed 0.975 compared to 42-45% for FRESCO-S. Loyola et al. (2018a) determined that OCRA cloud fractions retrieved over oceans are 0.1 unit more than those from retrievals like FRESCO-S that assume fixed cloud albedo. Differences over land are not as systematic and vary from negligible in the tropics and subtropics to > 0.1 unit more in the Arctic (Loyola et al., 2018a). The OCRA algorithm ordinarily includes red band reflectances, but

TROPOMI OCRA relies on initial cloud-free reflectances from OMI that excludes the red part of the visible spectrum, though its absence only induces a small negative cloud fraction bias of ~0.03 (Loyola et al., 2018a). ROCINN-CAL retrieves cloud optical thicknesses alongside cloud heights. These exceed 20 for most (84-93%) $1° \times 1°$ gridsquares used to cloud-slice TROPOMI, confirming that a cloud fraction threshold of 0.7 is sufficient to isolate optically thick clouds. The number of pixels in each cloud fraction threshold in Figure 6 suggests that a stricter cloud fraction threshold of 1.0 applied to the ROCINN-

CAL product might lead to a more consistent spatial distribution of UT $NO_2$ to that from FRESCO-S in Figure 5. The resultant ROCINN-CAL UT $NO_2$ using a cloud fraction threshold of 1.0 are in Figure S3. Half the number of cloud-sliced retrievals are obtained, as expected, and there are fewer retrievals over northern hemisphere high latitudes than in Figure 5. Those over the southern ocean in austral autumn and winter persist and may reflect enhanced occurrence of high-altitude clouds in these seasons over Antarctica (Verlinden et al., 2011). The average difference between ROCINN-CAL and FRESCO-S decreases



from 5-8% for the same cloud fraction threshold of 0.7 to 0.2-1.6% using a cloud fraction threshold of 1.0 for ROCINN-CAL
and 0.7 for FRESCO-S.

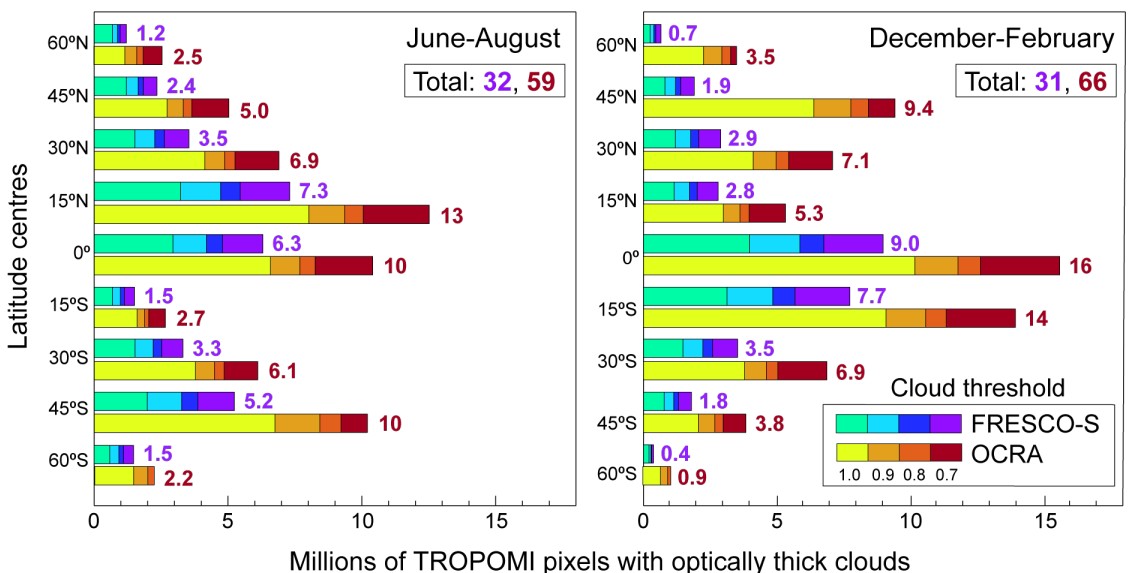

**Figure 6: Meridional distribution of FRESCO-S and OCRA optically thick clouds in the upper troposphere. Bars count the**
**occurrence of native TROPOMI pixels with cloud fractions ≥ 1.0, 0.9, 0.8, and 0.7 binned into 15° latitude bands in June-August**
**(left) and December-February (right) for FRESCO-S (cool colours) where FRESCO-S cloud top pressures are at 450-180 hPa, and**
**OCRA (warm colours) where ROCINN-CAL cloud top pressures are at 450-180 hPa. Values inset are latitude band and global total**
**number of TROPOMI pixels with cloud fraction ≥ 0.7.**

Figure 7 compares gridded cloud product cloud top pressures for June-August sampled where FRESCO-S cloud top pressures
are at 450-180 hPa and cloud fractions are at least 0.7. Cloud top pressures are spatially consistent in the tropics (R = 0.62 at
0-35°N, R = 0.85 at 0-35°N) and midlatitudes (R = 0.58 at 35-70°N, R = 0.63 at 35-70°S), but degrade north of 70°N (R =
0.31). Variability in cloud top pressures is similar for the two products in the tropics (regional mean standard deviation of 28-
33 hPa at 0-35°N and 30-31 hPa at 0-35°S), but deviates in the subtropics and midlatitudes (18 hPa for FRESCO-S, 24-30 hPa
for ROCINN-CAL) and more so in the Arctic (13 hPa for FRESCO-S, 54 hPa for ROCINN-CAL). There is no coincident data
south of 70°S in June-August. In December-February south of 70°S (Figure S4) there is a similarly weak correlation (R < 0.1)
and large difference in variability (19 hPa for FRESCO-S, 80 hPa for ROCINN-CAL). FRESCO-S does not account for
scattering within and below clouds and so estimates the height as the optical centroid of the cloud (Joiner et al., 2012). The
optical centroid is systematically lower in altitude (higher in pressure) than the physical cloud top, though FRESCO-S appears
to be more consistent with ground-based observations than ROCINN-CAL for high-altitude cloud top heights (Compernolle
et al., 2020). Loyola et al. (2018a) determined that cloud top altitudes from ROCINN-CAL were ~1 km (range: 0.6 km to >2
km) higher than those from a FRESCO-S type approach that assumes clouds are single layers with fixed albedo. Our test of
the effect of an artificial decrease in cloud top altitude of 1 km for cloud-slicing synthetic GEOS-Chem partial columns (Section



2) suggests that 1 km lower altitude cloud tops in FRESCO-S should lead to larger UT NO$_2$ than those from ROCINN-CAL, but the opposite is observed (Figure 5). This suggests that the effect of other differences between the cloud products on the

cloud-sliced UT NO$_2$ must dominate. Regression slopes in Figure 7 are less than unity, indicating that the difference in cloud top pressures between the two products decreases with pressure (increases with altitude). The implication for cloud-sliced UT NO$_2$ is greater global coverage with ROCINN-CAL, as clusters of TROPOMI pixels in the midlatitudes and polar regions overcome the 140 hPa cloud top pressure range threshold imposed in the cloud-slicing algorithm. In the tropics and subtropics, ROCINN-CAL has less cloud top pressure range than FRESCO-S for the same scenes. This leads to steeper cloud-slicing

regression slopes for ROCINN-CAL and explains the 4-9 pptv greater UT NO$_2$ than FRESCO-S in Figure 5.

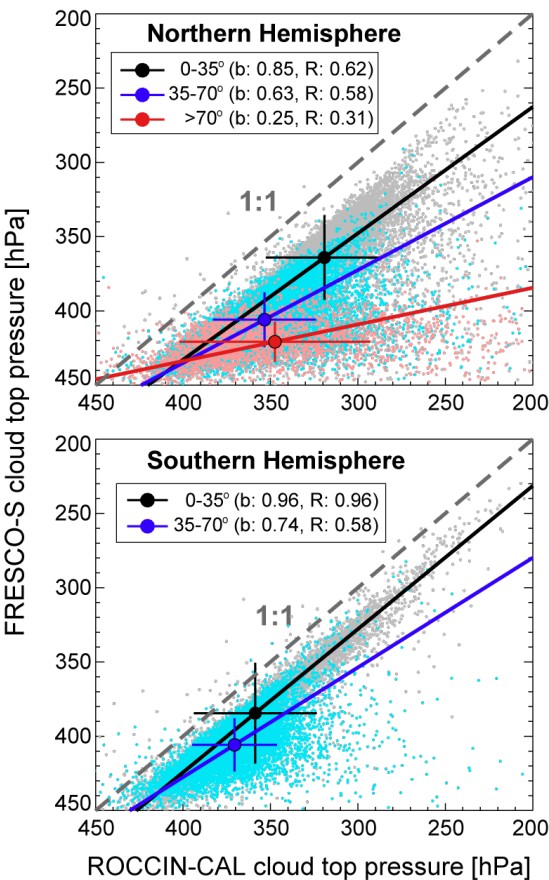

**Figure 7: Comparison of FRESCO-S and ROCINN-CAL cloud top pressures from optically thick clouds in the upper troposphere for June-August 2019. Data are gridded to 1° × 1° for TROPOMI pixels with FRESCO-S cloud fractions ≥ 0.7 and cloud top**
**pressures at 450-180 hPa. Small points are gridded seasonal means and lines are RMA regressions for the tropics (grey points, black regression line), subtropics and midlatitudes (cyan, blue), and the Arctic (pink, red). Large points are latitude band means and error bars are corresponding standard deviations. Grey dashed lines show the 1:1 relationship. Values in the legend are RMA regression slopes (b) and Pearson's correlation coefficients (R).**



## 5 Comparison of TROPOMI and OMI UT NO$_2$


We evaluate TROPOMI UT NO$_2$ with multiyear (2005-2007) seasonal mean cloud-sliced UT NO$_2$ from OMI at 5° × 8° (Marais et al., 2018). The OMI product is retrieved in a similar manner to TROPOMI, except that the GMI CTM is used to diagnose and remove steep gradients in NO$_2$ (≥ 0.33 pptv hPa$^{-1}$) and the OMI retrieval ceiling is lower (280 hPa, ~10 km) than TROPOMI (180 hPa, ~12.5 km). In regions where lightning is prevalent, the vertical distribution of NO$_2$ increases with altitude by 10-50 pptv across 280-180 hPa, as is observed with vertical profiles of NO$_2$ from spring-summer research aircraft campaign


measurements over the US (Boersma et al., 2011; Silvern et al., 2018). Strict filtering applied to cloud-slicing removes most scenes where the increase in NO$_2$ with altitude exceeds 33 pptv across 280-180 hPa, based on the synthetic experiment with GEOS-Chem. The influence of more than a 10-year gap between the OMI and TROPOMI UT NO$_2$ datasets on the comparison is challenging to quantify, due to paucity of routine measurements of NO$_2$ in the UT. The contribution of changes in lightning


activity should be small, as interannual variability is small (<5%) and there is no discernible trend in the long-term record of satellite observations of lightning flashes (Schumann and Huntrieser, 2007).

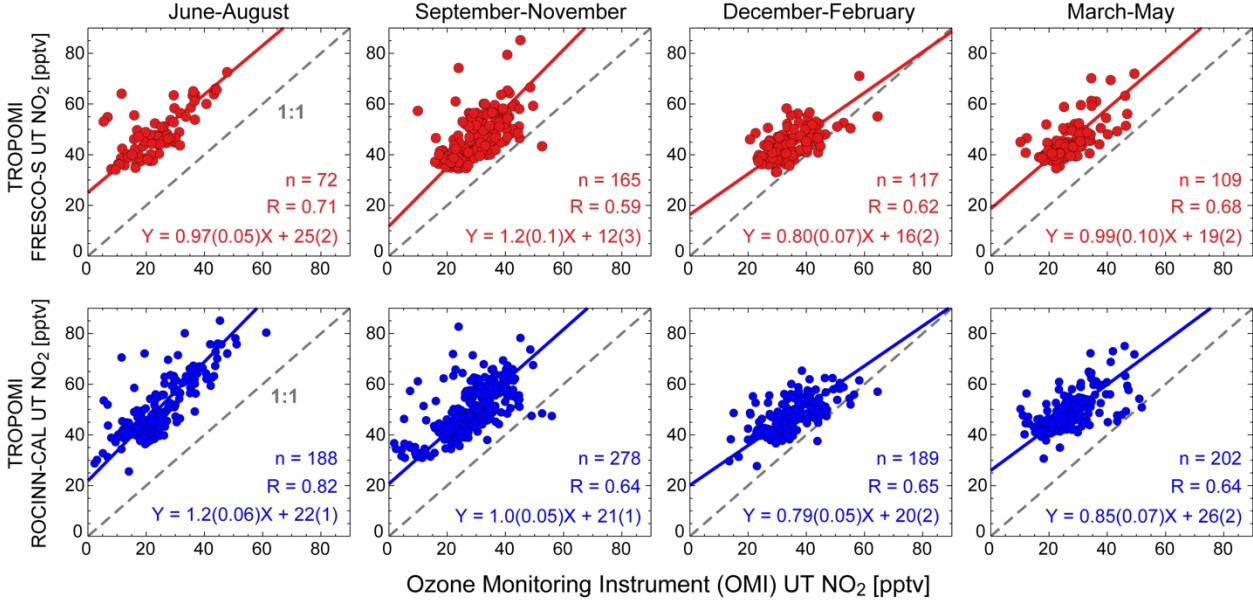

**Figure 8: Comparison of TROPOMI and OMI cloud-sliced UT NO$_2$. Points are seasonal means in June 2019 to May 2020 for**


**TROPOMI and January 2005 to December 2007 for OMI gridded to the same 5° × 8° (latitude × longitude) grid for FRESCO-S vs OMI (upper panel, red) and ROCINN-CAL vs OMI (lower panel, blue). Values give the number of points, Pearson's correlation coefficients (R), RMA regression coefficients, and, in parentheses, bootstrap resampling slope and intercept errors.**

Figure 8 evaluates spatial consistency between TROPOMI and OMI seasonal mean UT NO$_2$ on the OMI grid (5° × 8°) for TROPOMI cloud-sliced UT NO$_2$ 1° × 1° gridsquares with at least 10 cloud-sliced retrievals. TROPOMI is spatially consistent


with OMI in all seasons for both products (R = 0.6-0.8). The TROPOMI background is 12-25 pptv more than OMI for FRESCO-S and 20-26 pptv more than OMI for ROCINN-CAL, based on the intercepts in Figure 8. This may be due to the



observed increase in NO$_2$ with altitude across the sampling pressure ceilings of the two products (280 hPa for OMI, 180 hPa for TROPOMI). The OMI UT NO$_2$ product uses cloud information derived from OMI O$_2$-O$_2$ slant columns. The signal from the O$_2$-O$_2$ dimer declines with altitude, increasing uncertainty in the retrieval with altitude (Veefkind et al., 2016). High-altitude

clouds from OMI would have to be higher in altitude than TROPOMI to contribute to the positive offset in TROPOMI UT NO$_2$ in Figure 8, based on results from the synthetic test of lowering GEOS-Chem cloud top heights by 1 km. But the direction of the bias in OMI high-altitude cloud top heights compared to lidar-radar measurements does not appear to be systematic (Veefkind et al., 2016). The regression slopes in Figure 8 are closest to unity for June-August and March-May for FRESCO-S and September-November for ROCINN-CAL. The underestimate in variance in December-February in both products could

reflect the need to account for seasonality in the stratospheric variance correction. UT NO$_2$ obtained without applying correction factors to TROPOMI stratospheric and tropospheric columns (Figure S5) results in greater data density due to less variance in TROPOMI stratospheric columns, but the discrepancy with OMI is much greater. TROPOMI UT NO$_2$ background concentrations are 16-35 pptv more than OMI for FRESCO-S and 27-36 pptv more for ROCINN-CAL and slopes exceed unity in all seasons (1.3-1.7 for FRESCO-S, 1.2-1.5 for ROCINN-CAL).

**6 Conclusions**

We have developed new products of NO$_2$ in the upper troposphere (UT; ~8-12 km) by cloud-slicing partial columns of NO$_2$ from the space-based TROPOMI instrument. This involves regressing partial NO$_2$ columns against cloud top pressures and converting regression slopes to UT NO$_2$ mixing ratios.

We first refined and tested representativeness of cloud-sliced UT NO$_2$ by applying cloud-slicing to synthetic partial columns from the GEOS-Chem model. Synthetic cloud-sliced UT NO$_2$ are spatially consistent (R = 0.64) with the synthetic truth, but preferentially sampling cloudy scenes and substantial data loss lead to a resolution-dependent positive bias in cloud-sliced UT NO$_2$ of 11-22%.

Before applying cloud-slicing to TROPOMI, we evaluated TROPOMI with Pandora total columns at high-altitude sites (Mauna Loa, Izaña, Altzomoni) and Pandora and MAX-DOAS free tropospheric columns at Izaña. We identified discrepancies between TROPOMI and ground-based NO$_2$ measurements that include a 13% underestimate in TROPOMI stratospheric NO$_2$ variance and 50% overestimate in TROPOMI tropospheric columns.

We retrieved UT NO$_2$ from TROPOMI by applying the refined cloud-slicing algorithm to corrected TROPOMI partial columns above optically thick clouds with cloud top heights at 450-180 hPa using two alternate cloud products, FRESCO-S and ROCINN-CAL. ROCINN-CAL UT NO$_2$ has more extensive coverage (0°-70° N/S) than FRESCO-S (0°-45° N/S) due to its greater abundance of optically thick clouds. Coincident UT NO$_2$ from the two products exhibit similar spatial distribution, but

background UT $NO_2$ from ROCINN-CAL is 4-9 pptv more than FRESCO-S. This is due to steeper cloud-slicing regression slopes for ROCINN-CAL, as cloud top heights between the two products deviate with increasing cloud top pressure. Ongoing validation studies are needed to resolve these differences.

Both products are spatially correlated with the existing coarse resolution (5° latitude × 8° longitude) Ozone Monitoring Instrument (OMI) product, except that TROPOMI is 16-36 pptv more than OMI that we reason is due to the widely documented

increase in $NO_2$ with altitude from the OMI pressure ceiling (280 hPa) to that for TROPOMI (180 hPa), but signal saturation of TROPOMI pixels leading to blooming over bright high-altitude clouds could also contribute.

TROPOMI UT $NO_2$ products presented here have the potential to provide routine, extensive and consistent measurements of $NO_x$ in the UT and, as TROPOMI observations accumulate, aid in characterising interannual and long-term variability in $NO_x$

in the under sampled UT.

**Code Availability**

Python code used to process TROPOMI data is available at https://doi.org/10.5281/zenodo.4058442.

**Data Availability**

Data can be requested from EAM for TROPOMI UT $NO_2$, SC for OMI UT $NO_2$, MN-C for MAX-DOAS slant columns, and

RR for MAX-DOAS vertical columns. Python code used to process TROPOMI data is available at https://doi.org/10.5281/zenodo.4058442.

**Author Contributions**

EAM led the analysis and writing, simulated GEOS-Chem, and cloud-sliced TROPOMI UT $NO_2$ and GEOS-Chem. JFR refined and documented python processing code. RGR retrieved MAX-DOAS vertical tropospheric columns. HE and FB

provided guidance on best use and evaluation of TROPOMI $NO_2$. SC and JJ retrieved OMI UT $NO_2$ and provided guidance on cloud-slicing. NA and AR are Pandora site PIs, and AC is PI of PGN. MNC maintains the RASAS-II instrument and retrieved the MAX-DOAS slant columns at Izaña. LG conducted MAX-DOAS geospatial sampling sensitivity tests for Izaña.

**Competing Interests**

The authors declare that they have no conflict of interest.



### Acknowledgements

The project received funding from the European Research Council under the European Union's Horizon 2020 research and innovation programme (through a Starting Grant awarded to EAM, grant number 851854_UPTROP). The authors are grateful to Piet Stammes and colleagues for retrieval of the FRESCO-S cloud product and Diego Loyola and colleagues for retrieval of the OCRA and ROCINN-CAL cloud product. The PGN is a bilateral project supported with funding from NASA and ESA. The Pandora measurements in Altzomoni were possible thanks to financial support of Conacyt-AEM (Grant No. 275239) and technical assistance from Alejandro Bezanilla. RR acknowledges funding from the Australian Research Council Centre of Excellence for Climate Extremes and the Dr Albert Shimmins Memorial Fund through the University of Melbourne.

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
