# Peer review of "New Observations of Upper Tropospheric NO2 from TROPOMI"

_Atmospheric Measurement Techniques, 2020_

## Referee Comment (RC1) · Anonymous Referee #1 · 21 Oct 2020

Overview: This is a solid contribution describing a novel upper tropospheric NO2 product derived by cloud-slicing TROPOMI retrievals.

General Comments:

L406: What do you mean by corrected TROPOMI columns. Are you referring to the 13% change in stratospheric variance and 50% decrease in tropospheric columns referred to in line 386. If yes, were these large adjustments based on a comparison with just 3 sites? Justify.

L417: What are the pros and cons of producing two UT NO2 products as opposed to a merged product.

L450: What is the spatial correlation between these distributions and the OTDLIS cli-

matology?

L593 (and in abstract) Can these estimates of UT NO2 be compared directly to aircraft measurements as hinted at in the abstract or do they need to be averaged over weeks or seasons before such a comparison is meaningful. The near zero correlation of with daily free-troposphere values seems to suggest the latter.

How much value would be added if this product was created using multiple years of TROPOMI data as opposed to one year?

Specific Comments L27: Consider deleting the discussion of synthetic columns from the abstract. It may discourage the general reader from reading the manuscript. i.e., consider deleting the section beginning with "This follows refinement" . . . and ending with "overlying stratospheric NO2".

L79: May want to mention SAGE III, which has a few UT measurements of NO2 L130-142. Many details here. My following questions are an attempt to see if I understand what you did.

L132: What is a cluster? Is it the collection of 0.25 x 0.3125 degree boxes assigned to a 4 x 5 box? If yes, are you saying that you throw out an entire cluster if the stratospheric column NO2 relative standard deviation exceeds 0.02?

L136: How are scenes determined? If you break clusters into multiple scenes why don't you start doing this at 80 (2 x 40) as opposed to 100?

L141: How far into a deep convective cloud can TROPOMI see? Does the signal penetrate a few km below the top edge of the highest model layer?

L145: What do you mean by "error on the slope"? Is this the uncertainty of the slope?

L147: I don't understand what you mean by "negative slopes". Wouldn't you expect the partial column to decrease as the cloud top height increases because higher heights mean that TROPOMI is sampling less of the troposphere?

L148: "Converted to mixing ratio". Does this mean that you are dividing by the mass of air between the cloud top pressure and the tropopause?

L179: What is the source of this stratospheric variability? Is it aircraft and/or lightning-NOx?

L202: For TROPOMI, do you use the effective cloud fraction or the effective cloud fraction in the NO2 window?

L216: What do you mean by "free tropospheric" NO2 from TROPOMI. Is this what you called "upper tropospheric" NO2 earlier?

L278: Is AMFtrop.geo a standard TROPOMI data product or did you derive this variable from other terms (VZA, SZA etc.)? Ok, it looks like it is non-standard but easily calculated as you show later.

L349: Could you be more specific here. How was the stratospheric NO2 variance correction applied?

L360: Are estimates of free-tropospheric NO2 from TROPOMI possible on all days or only on days with variations in cloud-cover over a 4 x 5 grid. How many data points are there for each instrument?

L371: Could you explain the meaning of "the sampling footprint shifts by at least 2 degrees in latitude between . . . solstices. i.e., When is it larger

L421: The standalone product version number . . . I would move this sentence later in the paragraph and start with "Cloud fractions from the standalone product . . ."

Technical Corrections. L73: but these are prevalent over heavily — but these are mostly limited to heavily

L74: and are often — that are often

L115: What does "last accessed" mean? Is this the last day that you checked the link

or is it the (last) day that downloaded the model version?

L135: likely — possible

L206: can be ∼1 km — are ∼ 1 km

L241: downscaling — decreasing

L307: downscaling — downward adjustment

L381: downscale — decrease (Downscaling refers to a change in spatial resolution)

L413: 2018)) — 2018)
* * *

---

## Referee Comment (RC2) · Anonymous Referee #2 · 8 Jan 2021

**Review of "New Observations of Upper Tropospheric NO$_2$ from TROPOMI"**

**January 8, 2021**

Marais et al. "New Observations of Upper Tropospheric NO$_2$ from TROPOMI" presents an application of their cloud slicing technique previously used with OMI NO$_2$ data to TROPOMI NO$_2$ data. The high resolution of TROPOMI pixels allows the cloud slicing approach to be applied to much finer resolutions, comparable to typical global chemical transport model resolutions. This will be valuable to study the representation of upper tropospheric (UT) processes in these models since, as the authors say, in situ UT observations are sparse.

This manuscript should be published in AMT with minor revisions, mostly for clarity. I will address my comments by section below. I do deeply appreciate the authors providing their code in a proper repo!

**Sect 2: Cloud slicing GEOS-Chem**

- Line 132: "These are then screened to remove clusters with non-uniform GEOS-Chem stratospheric NO2.... Additional filtering is applied to clusters to remove extreme NO2 partial columns...." I don't see any comparable filters applied to TROPOMI data. Have you applied such filters, and if not, how does the GEOS-Chem test look without them? My concern is that having additional filters on the GEOS-Chem data not possible on the TROPOMI data limits its utility as a synthetic test.

- Line 148: Recommend citing Ziemke et al. 2001 here for the conversion from slope to mixing ratio.

- Line 151: Could you give more detail on what you mean by "Gaussian weighting individual estimates of cloud-sliced UT NO2 to the pressure centre"?

- Line 170-175: Do you have any hypothesis for why the size of the binning grid squares affects the magnitude of the cloud-sliced NO2, both here and with the TROPOMI data in Sect 4? This kind of systematic difference from resolution raises the question of which resolution is the most accurate, so an understanding of what drives that behavior would help.

- Line 202: "The cloud fraction retrieved from TROPOMI is an effective or radiometric cloud fraction that is systematically **less** than the physical cloud fraction from the

model." Is "less" correct? Usually radiative cloud fractions are greater than geometric cloud fractions because they are weighted by the amount of light coming from the clear vs. cloudy parts of the scene, and since cloud tops are highly reflective, this boosts the radiance cloud fraction (Bucsela et al. 2013, p. 2611 below Eq. 5; Laughner et al. 2018 Fig. S3c).

- Line 210: "This is because the decrease in cloud top altitude leads to a larger increase in the vertical extent of partial columns above high-altitude clouds than those above low-altitude clouds leading to steeper regression slopes and larger UT NO2." This is a bit confusing—did you mean that a 1 km difference is a larger *relative* increase in column for high altitude clouds than low altitude, since there is less column above high altitude clouds?

**Sect 3: TROPOMI evaluation**

- Around line 300: are you comparing clear-sky and cloudy-sky TROPOMI VCDs calculated with the geometric-only AMF to Pandora, or just cloudy TROPOMI pixels where the cloud top height is similar to the elevation of these Pandoras? If the former, I'm not sure I understand the reason to include this comparison. The AMF used here relies on assumptions that may hold above clouds, but become more uncertain when the total column is considered.

- Line 328: "The underestimate in stratospheric NO2 variance would lead to an overestimate in the relative contribution of the stratosphere to the total column for small column densities and vice versa." A few points:

  - I'm not convinced by this. An underestimation in the variance doesn't indicate when it is under- or over- estimated. I think I understand what you are implying - that e.g. when the total column decreased by 50% the stratosphere *typically* changes by less than that, and if we assume that the stratosphere is proportional to the total column, then your argument holds. But this would be easier to demonstrate by comparing the stratospheric columns directly (e.g. scatter plot of TROPOMI strat. vs. Pandora strat.) rather than invoking variability.

  - Given that the stratosphere is assumed to be fairly uniform over moderate spatial scales, I'm not automatically convinced that even a 1x1 deg model significantly smoothes the stratosphere. Dirksen et al. (2011) for example showed that the OMI DOMINO stratospheric product captured a polar vortex event reasonably well by comparison with ground based data.

  - One final point I am concerned about is the difference in stratospheric dynamics over Mauna Loa and the rest of the tropics compared with the extratropics due to the Brewer-Dobson circulation. Have you considered the Pandora vs. TROPOMI stratospheric columns at, for example, Fort McKay, Fairbanks, Eureka, or Ny-Alesund to see if this same error in the stratosphere occurs in the mid- and high-latitudes?

- Line 341 (whole paragraph)

1. Is this really relevant to this paper? It seems focused on general TROPOMI validation, not cloud slicing.

2. If it is relevant, I'm not convinced that the stratospheric variance is the largest reason for these urban/rural differences. Underestimation of satellite $NO_2$ in urban areas and overestimation outside them is a common result of coarse a priori profiles that are too coarse (Russell et al., 2011) because the prior profile is a mix of urban & rural characteristics. Since Pandora is substantially less sensitive to the vertical profile of $NO_2$, it follows that comparisons with Pandora would illuminate this difference.

- Line 349: "After applying the stratospheric NO2 variance correction"— please explain how this correction is applied.

- Line 364: "We impose a modest threshold" - does this mean only TROPOMI columns >4e13 are used? TROPOMI columns <4e13 are set to 4e13? Something else?

- Line 375: "TROPOMI free tropospheric columns (red crosses in Figure 4)" - I am a little confused here. Has this section been discussing TROPOMI observations with just the free tropospheric (excluding boundary layer) or the total tropospheric columns? From Eqs. 1 & 2, there is no indication that the boundary layer has been removed, just the tropospheric VCD was recalculated with a geometry only AMF.

**Sect 4: UT NO2 retrieval**

- Line 416, paragraph: it is confusing to use "The standalone product version number changes from 01-01-07 to 01-01-08..." as the only transition from describing the FRESCO cloud product to describing the offline cloud product. This is aggravated because, earlier in the paper, FRESCO and ROCINN were used as the names of the products rather than "in the NO2 files" and "the standalone product". Also "These are FRESCO-S from the same data file as TROPOMI NO2 and the standalone offline (OFFL) cloud product," is ambiguous as to whether it means one product is FRESCO-S from the NO2 files and the other is the standalone product *or* FRESCO-S is the product in both the NO2 and standalone files. This paragraph could be made clearer.

- Moving the descriptions of the retrievals to the supplement (or removing altogether) could streamline the paper and focus on the cloud slicing aspects.

  – It seems that the most important difference between these products is whether they treat clouds as reflective surfaces or 3D objects. Focusing this paragraph on that, rather than the details of how these algorithms work, would likely help readability.

- Why is there so much more data in the northern high latitudes in Sept-Nov than Jun-Aug (Fig 5)? It seems like the greater summer illumination would make it easier to get data in Jun-Aug. According to Figs. 6 and S2, while there are more clouds in Sep-Nov than Jun-Aug, there are still 1.2 million cloudy pixels in Jun-Aug.

- Line 459: "Coincident gridsquares of the two data sets..." Since the previous sentence was talking about anthropogenic contamination, it isn't immediately clear that the "two data sets" are the two cloud sliced ones. Perhaps a paragraph break would help?

- Line 518: "...as clusters of TROPOMI pixels in the midlatitudes and polar regions overcome the 140 hPa cloud top pressure range..." a bit awkwardly phrased, as "overcome" somewhat implies more clouds exceeding the 140 hPa cutoff, which would imply fewer, not more clusters passing the filter.

**Sect 5: Comparison between TROPOMI and OMI**

- Line 531: Was 2005-2007 (rather than a multiyear period overlapping TROPOMI) chosen because that was before the row anomaly began degrading OMI retrievals? If so, would be good to say so to explain why this time period was chosen even though the impact of the 10+ year gap is difficult to quantify.

- Line 540: Conversely, Romps et al. (2014) suggest lightning will increase 12%/°C. Given that Schumann and Huntrieser published at the beginning of this 10 year gap, a more recent reference would strengthen the claim that lightning has not increased.

- Line 559: "The underestimate in variance in December-February in both products could reflect the need to account for seasonality in the stratospheric variance correction." Cannot tell that the variance is underestimated in both from Fig. 8 since it is just comparing the two satellites' values, without a truth metric. Please indicate how you know both are underestimated.

- Line 560: "UT NO2 obtained without applying correction factors to TROPOMI stratospheric and tropospheric columns (Figure S5) results in greater data density due to less variance in TROPOMI stratospheric columns, but the discrepancy with OMI is much greater." This raises the question of why TROPOMI needed the 50% reduction in tropospheric columns. Since the TROPOMI:OMI slopes increase, that suggests (assuming that 2005-2007 and 2019-2020 are comparable) that TROPOMI has a multiplicative bias vs. OMI. Which OMI product (DOMINO v1/v2, NASA SP v1/v2/v3) are you using in this comparison? The QA4ECV product is probably the most similar to the TROPOMI retrieval; in the past, DOMINO has been clearly higher than NASA SP (e.g. Hains et al., 2010), so if the OMI 2005-2007 product uses NASA, then it's possible this differences are retrieval driven (though that comparison is muddled since you're not using the standard tropospheric AMF).

**References**

Bucsela, E. J., Krotkov, N. A., Celarier, E. A., Lamsal, L. N., Swartz, W. H., Bhartia, P. K., Boersma, K. F., Veefkind, J. P., Gleason, J. F., and Pickering, K. E.: A new stratospheric and tropospheric NO$_2$ retrieval algorithm for nadir-viewing satellite instruments: applications to OMI, Atmospheric Measurement Techniques, 6, 2607–2626, doi:10.5194/amt-6-2607-2013, URL https://amt.copernicus.org/articles/6/2607/2013/, 2013.

Dirksen, R. J., Boersma, K. F., Eskes, H. J., Ionov, D. V., Bucsela, E. J., Levelt, P. F., and Kelder, H. M.: Evaluation of stratospheric NO2 retrieved from the Ozone Monitoring Instrument: Intercomparison, diurnal cycle, and trending, Journal of Geophysical Research: Atmospheres, 116, doi:https://doi.org/10.1029/2010JD014943, URL https://agupubs.onlinelibrary.wiley.com/doi/abs/10.1029/2010JD014943, 2011.

Hains, J. C., Boersma, K. F., Kroon, M., Dirksen, R. J., Cohen, R. C., Perring, A. E., Bucsela, E., Volten, H., Swart, D. P. J., Richter, A., Wittrock, F., Schoenhardt, A., Wagner, T., Ibrahim, O. W., van Roozendael, M., Pinardi, G., Gleason, J. F., Veefkind, J. P., and Levelt, P.: Testing and improving OMI DOMINO tropospheric NO2 using observations from the DANDELIONS and INTEX-B validation campaigns, Journal of Geophysical Research: Atmospheres, 115, doi:https://doi.org/10.1029/2009JD012399, URL https://agupubs.onlinelibrary.wiley.com/doi/abs/10.1029/2009JD012399, 2010.

Laughner, J. L., Zhu, Q., and Cohen, R. C.: The Berkeley High Resolution Tropospheric $NO_2$ product, Earth System Science Data, 10, 2069–2095, doi:10.5194/essd-10-2069-2018, URL https://essd.copernicus.org/articles/10/2069/2018/, 2018.

McKenna, D. S., Konopka, P., Grooß, J.-U., Günther, G., Müller, R., Spang, R., Offermann, D., and Orsolini, Y.: A new Chemical Lagrangian Model of the Stratosphere (CLaMS) 1. Formulation of advection and mixing, Journal of Geophysical Research: Atmospheres, 107, ACH 15–1–ACH 15–15, doi:https://doi.org/10.1029/2000JD000114, URL https://agupubs.onlinelibrary.wiley.com/doi/abs/10.1029/2000JD000114, 2002.

Romps, D. M., Seeley, J. T., Vollaro, D., and Molinari, J.: Projected increase in lightning strikes in the United States due to global warming, Science, 346, 851–854, doi:10.1126/science.1259100, URL https://science.sciencemag.org/content/346/6211/851, 2014.

Russell, A. R., Perring, A. E., Valin, L. C., Bucsela, E. J., Browne, E. C., Wooldridge, P. J., and Cohen, R. C.: A high spatial resolution retrieval of $NO_2$ column densities from OMI: method and evaluation, Atmospheric Chemistry and Physics, 11, 8543–8554, doi:10.5194/acp-11-8543-2011, URL https://acp.copernicus.org/articles/11/8543/2011/, 2011.

---

## Author Comment (AC1) · 8 Feb 2021

Please find attached the point-by-point responses to reviewer comments and the manuscript with changes tracked.

Please also note the supplement to this comment:
https://amt.copernicus.org/preprints/amt-2020-399/amt-2020-399-AC1-supplement.zip

---

## Author Comment (AC2) · 8 Feb 2021

Please find attached a zipped file of two documents: (1) point-by-point responses to reviewer comments, and (2) the manuscript with changes tracked.

Please also note the supplement to this comment:
https://amt.copernicus.org/preprints/amt-2020-399/amt-2020-399-AC2-supplement.zip

---

## Author Response (AR2)

**RESPONSES TO REVIEWERS**

Ms. Ref. No.: Atmos. Meas. Techn. Discuss., doi:10.5194/amt-2020-399.

Title: New Observations of Upper Tropospheric $NO_2$ from TROPOMI

Reviewer comments are in blue, responses are in black Line numbers in the blue text correspond to the submitted manuscript and line numbers in black correspond to the attached PDF with all changes in response to reviewer comments tracked. Line numbers are for the original tracked changes manuscript submitted to the editor on 8 Feb 2021.

Additional minor technical corrections following submission of our responses to reviewer comments are for the updated tracked changes manuscript submitted to the editor on 13 Feb 2021. These are indicated in Italics below.

**Notes to Editor:**

The version names of the Pandora total and tropospheric columns mixed and have updated these in the manuscript (line 294).

We have also addressed technical issues identified after addressing reviewer comments:
1. URLs and DOIs for the data sets and model are in the reference list and the in-text citations have been updated for GEOS-Chem (*line 125*), the Pandonia Global Network (*line 295*), and the S5P Data Hub (*line 335*) LIS-OTD (*lines 648-649*).
2. TROPOMI retrieval approach is now referred to as DOMINO rather than DOAS (*line 779*).

The correct revised manuscript should have also been uploaded this time. I now realise the error was because it wasn't clear whether the manuscript requested was the original or the revised without changes tracked, as the upload instructions did not specify this.

**Responses to Anonymous Referee #1:**

*Overview: This is a solid contribution describing a novel upper tropospheric NO2 product derived by cloud-slicing TROPOMI retrievals.*

General Comments:

*L406: What do you mean by corrected TROPOMI columns. Are you referring to the 13% change in stratospheric variance and 50% decrease in tropospheric columns referred to in line 386. If yes, were these large adjustments based on a comparison with just 3 sites? Justify.*

We now elaborate on the details of the correction applied to TROPOMI (lines 542-543). Yes, the corrections applied to TROPOMI are based on these 3 sites. We state this too now in this paragraph so that this is clear (line 543) and also reconfirm that the correction factor for the stratospheric variance is for a single site (lines 779-780). We already confirm that UT $NO_2$ obtained after applying these correction factors is more consistent with OMI than cloud-sliced UT $NO_2$ obtained without these correction factors (lines 780-781).

*L417: What are the pros and cons of producing two UT NO2 products as opposed to a merged product.*

As the two datasets aren't independent, we assume the reviewer is referring to a merged product that would use one of the datasets as the baseline (FRESCO-S, perhaps) and filling in data gaps with the other (ROCINN-CAL) product. We suspect that this would lead to a product with the same coverage as ROCINN-CAL and would mix two products that we show use cloud products with systematic differences (lines 553-568).

*L450: What is the spatial correlation between these distributions and the OTDLIS climatology?.*

We now summarize spatial consistency and regression statistics obtained from comparison of TROPOMI UT NO$_2$ and the OTD-LIS climatology (lines 641-648).

*L593 (and in abstract) Can these estimates of UT NO2 be compared directly to aircraft measurements as hinted at in the abstract or do they need to be averaged over weeks or seasons before such a comparison is meaningful. The near zero correlation of with daily free-troposphere values seems to suggest the latter.*

We found in a previous assessment of OMI UT NO$_2$ with NASA DC8 aircraft observations over North America that averaging over multiple campaigns and months (March-September) was required (Marais et al., 2018), so we suspect the same would be the case for TROPOMI due to sparsity of data in the upper troposphere from these aircraft campaigns.

Specific Comments:

*L27: Consider deleting the discussion of synthetic columns from the abstract. It may discourage the general reader from reading the manuscript. i.e., consider deleting the section beginning with "This follows refinement" … and ending with "overlying stratospheric NO2".*

We now only state in the abstract that we refine the cloud-slicing algorithm with synthetic columns from GEOS-Chem (line 29).

*L79: May want to mention SAGE III, which has a few UT measurements of NO2*

We now include two relevant SAGE-III NO$_2$ references: Brogniez et al. (2002) and Dubé et al. (2020) (lines 87-88) (highlighted in the attached PDF)

*L130-142. Many details here. My following questions are an attempt to see if I understand what you did.*

Thank you for your questions. These have led to a clearer description of the approach.

*L132: What is a cluster? Is it the collection of 0.25 x 0.3125 degree boxes assigned to a 4 x 5 box? If yes, are you saying that you throw out an entire cluster if the stratospheric column NO2 relative standard deviation exceeds 0.02?*

Yes, we discard the entire cluster. We have reworded this for clarity (lines 142-143).

*L136: How are scenes determined? If you break clusters into multiple scenes why don't you start doing this at 80 (2 x 40) as opposed to 100?*

We have changed "scenes" to "clusters" for clarity (line 158). We initially chose 100 over 80 to be conservative in our selection of scenes to subdivide, as this is a new addition to the cloud-slicing algorithm. We conducted a test with the synthetic partial columns using 80 instead of 100 and find that this only leads to a small difference in seasonal mean UT $NO_2$ and the number of successful retrievals. We now say that a threshold of 80 or 100 could be used (lines 158-159).

*L141: How far into a deep convective cloud can TROPOMI see? Does the signal penetrate a few km below the top edge of the highest model layer?*

This depends on the opacity of the cloud. When we cloud-slice TROPOMI, we only consider scenes above optically thick clouds to reduce transmission of light through the clouds and contamination of the partial column of $NO_2$ with $NO_2$ below the cloud. We have stated this in a few locations in the manuscript already (lines 248 and 549). In line 248 we provide additional explanatory text for clarity.

*L145: What do you mean by "error on the slope"? Is this the uncertainty of the slope?*

We have reworded this to clarify that the error on the slope is estimated as the standard deviation from bootstrap resampling (lines 168-169).

*L147: I don't understand what you mean by "negative slopes". Wouldn't you expect the partial column to decrease as the cloud top height increases because higher heights mean that TROPOMI is sampling less of the troposphere?*

We looked into the spatial distribution of these negative slopes further to find that these mostly occur over remote regions where $NO_2$ concentrations are low that would lead to poor dynamic range in the regression. We now specify that these are mostly over remote locations (lines 169-171).

*L148: "Converted to mixing ratio". Does this mean that you are dividing by the mass of air between the cloud top pressure and the tropopause?*

We now point readers to the Ziemke et al. (2001) paper where the derivation of this conversion is detailed (lines 172-173).

*L179: What is the source of this stratospheric variability? Is it aircraft and/or lightning-NOx?*

We now state that our threshold for stratospheric variability is very strict to make apparent that it is susceptible to small changes in stratospheric $NO_2$ (line 143) and also state that $NO_2$ formation in the stratosphere is mostly from oxidation of $N_2O$ in the middle stratosphere and that its variability is dominated by changes in solar insolation and circulation dynamics (lines 144-146).

*L202: For TROPOMI, do you use the effective cloud fraction or the effective cloud fraction in the NO2 window?*

As stated in the text (lines 554-555), we use FRESCO-S effective cloud fractions obtained at 758-766 nm (the $O_2$ A-band).

*L216: What do you mean by "free tropospheric" NO2 from TROPOMI. Is this what you called "upper tropospheric" NO2 earlier?*

We have reworded this to clarify that the free troposphere that these sites sample include $NO_2$ from the upper and middle troposphere (lines 277-280).

*L278: Is AMFtrop.geo a standard TROPOMI data product or did you derive this variable from other terms (VZA, SZA etc.)? Ok, it looks like it is non-standard but easily calculated as you show later.*

We now make clear that we calculate $AMF_{trop,geo}$ (line 362).

*L349: Could you be more specific here. How was the stratospheric NO2 variance correction applied?*

We now elaborate on the stratospheric variance correction calculation (lines 424-426).

*L360: Are estimates of free-tropospheric NO2 from TROPOMI possible on all days or only on days with variations in cloud-cover over a 4 x 5 grid. How many data points are there for each instrument?*

TROPOMI daily mean free tropospheric $NO_2$ column densities are obtained on almost all days in the year (362 out of 366) for the coincident criteria specified in the text, that is, ~20 km from the ground monitoring site (line 370). We now include in Figure 4 the number of daily means for each instrument (221 for Pandora, 173 for MAX-DOAS, and 362 for TROPOMI).

*L371: Could you explain the meaning of "the sampling footprint shifts by at least 2 degrees in latitude between … solstices. i.e., When is it larger*

We have reworded this for clarity (lines 480).

*L421: The standalone product version number … I would move this sentence later in the paragraph and start with "Cloud fractions from the standalone product …"*

This sentence has been removed to address reviewer #2's comment that this paragraph should be shortened to only include pertinent information to improve the flow of this section (lines 553-568).

Technical Comments:

*L73: but these are prevalent over heavily—but these are mostly limited to heavily*

Changed (line 81).

*L74: and are often — that are often*

Changed (line 82).

*L115: What does "last accessed" mean? Is this the last day that you checked the link or is it the (last) day that downloaded the model version?*

Changed to "accessed" to clarify that this is the date the model code was obtained (line 125)

*L135: likely — possible*
Rewritten to "As many as 256 0.25° × 0.3125° partial columns can be gathered in a 4° × 5° grid" (line 147 on p. 4 and line 157 on p. 5).

*L206: can be ~1 km — are ~ 1 km*
All other usages of "~" do not include a space between "~" and the number, so for consistency this is unchanged.

*L241: downscaling — decreasing*
Changed (line 307).

*L307: downscaling — downward adjustment*
Changed (line 384).

*L381: downscale — decrease (Downscaling refers to a change in spatial resolution)*
Changed (line 506).

*L413: 2018)) — 2018)*
Double brackets are correct, but to avoid confusion, we've changed brackets to commas (lines 548-549).
* * *
**Responses to Anonymous Referee #2:**

*Marais et al. "New Observations of Upper Tropospheric NO2 from TROPOMI" presents an application of their cloud slicing technique previously used with OMI NO2 data to TROPOMI NO2 data. The high resolution of TROPOMI pixels allows the cloud slicing approach to be applied to much finer resolutions, comparable to typical global chemical transport model resolutions. This will be valuable to study the representation of upper tropospheric (UT) processes in these models since, as the authors say, in situ UT observations are sparse.*

*This manuscript should be published in AMT with minor revisions, mostly for clarity. I will address my comments by section below. I do deeply appreciate the authors providing their code in a proper repo!*

*Sect 2: Cloud slicing GEOS-Chem*
*Line 132: "These are then screened to remove clusters with non-uniform GEOS-Chem stratospheric NO2.... Additional filtering is applied to clusters to remove extreme NO2 partial columns...." I don't see any comparable filters applied to TROPOMI data. Have you applied such filters, and if not, how does the GEOS-Chem test look without them? My concern is that having additional filters on the GEOS-Chem data not possible on the TROPOMI data limits its utility as a synthetic test.*
The same filtering steps are applied to GEOS-Chem and TROPOMI. We now include "data filtering" to clarify that the method used to cloud-slice TROPOMI is consistent to that applied to the model (line 537).

*Line 148: Recommend citing Ziemke et al. 2001 here for the conversion from slope to mixing ratio.*
Added (line 94).

*Line 151: Could you give more detail on what you mean by "Gaussian weighting individual estimates of cloud-sliced UT NO2 to the pressure centre"?*
We now provide the equation for the Gaussian weights (lines 177-179).

*Line 170-175: Do you have any hypothesis for why the size of the binning grid squares affects the magnitude of the cloud-sliced NO2, both here and with the TROPOMI data in Sect 4? This kind of systematic difference from resolution raises the question of which resolution is the most accurate, so an understanding of what drives that behavior would help.*
We find that a larger relative proportion of clusters with absolute $NO_2$ vertical gradient $> 0.33$ pptv $hPa^{-1}$ are retained at coarser resolution. We now include this explanation in the manuscript (lines 216-219) and, in the previous paragraph, also give the reason for removing scenes with $NO_2$ with steep vertical gradients (lines 185-186).

*Line 202: "The cloud fraction retrieved from TROPOMI is an effective or radiometric cloud fraction that is systematically less than the physical cloud fraction from the model." Is "less" correct? Usually radiative cloud fractions are greater than geometric cloud fractions because they are weighted by the amount of light coming from the clear vs. cloudy parts of the scene, and since cloud tops are highly reflective, this boosts the radiance cloud fraction (Bucsela et al. 2013, p. 2611 below Eq. 5; Laughner et al. 2018 Fig. S3c).*
Thank you for identifying this error. This has been corrected and the references you provided cited (lines 252 and 669). As the difference between the geometric and radiance cloud fractions decrease with increase in cloud fraction, we keep the test the same, that is, cloud-slicing applied to GEOS-Chem synthetic columns over clouds with geometric cloud fraction $> 0.7$.

*Line 210: "This is because the decrease in cloud top altitude leads to a larger increase in the vertical extent of partial columns above high-altitude clouds than those above low-altitude clouds leading to steeper regression slopes and larger UT NO2." This is a bit confusing – did you mean that a 1 km difference is a larger relative increase in column for high altitude clouds than low altitude, since there is less column above high altitude clouds?*
Yes, that's correct. Thank you for the suggested rephrasing. We have amended the text (lines 260-261).

*Sect 3: TROPOMI evaluation*
To address many of the related comments for this section, we now explicitly state "free troposphere" rather than just "troposphere", where relevant, to ensure it is clear that the assessment is for the portion of the troposphere that excludes the boundary layer.

*Around line 300: are you comparing clear-sky and cloudy-sky TROPOMI VCDs calculated with the geometric-only AMF to Pandora, or just cloudy TROPOMI pixels where the cloud top height is similar*

*to the elevation of these Pandoras? If the former, I'm not sure I understand the reason to include this comparison. The AMF used here relies on assumptions that may hold above clouds, but become more uncertain when the total column is considered.*

We now clarify that this is a comparison of clear-sky Pandora and would effectively be all-sky for TROPOMI, due to its larger sampling footprint (lines 372-376). This comparison is crucial, as it helps us identify any issues with the TROPOMI stratospheric and free tropospheric columns that would impact the cloud-sliced UT $NO_2$. Without this comparison, TROPOMI UT $NO_2$ is much greater than TROPOMI UT $NO_2$ without this correction (Figure S5) and much greater than the OMI UT $NO_2$ product (Figure 8). We already discuss the impact of stratospheric variance issues on cloud-sliced UT $NO_2$ (lines 413-414) and now also state the impact of a positive offset in free tropospheric columns on cloud-sliced UT $NO_2$ (line 491 on p. 14 and line 506 on p. 15)

*Line 328: "The underestimate in stratospheric NO2 variance would lead to an overestimate in the relative contribution of the stratosphere to the total column for small column densities and vice versa." A few points:*

- *I'm not convinced by this. An underestimation in the variance doesn't indicate when it is under- or over- estimated. I think I understand what you are implying - that e.g. when the total column decreased by 50% the stratosphere typically changes by less than that, and if we assume that the stratosphere is proportional to the total column, then your argument holds. But this would be easier to demonstrate by comparing the stratospheric columns directly (e.g. scatter plot of TROPOMI strat. vs. Pandora strat.) rather than invoking variability.*

The comparison of Pandora and TROPOMI total columns at Mauna Loa serves as an assessment of the stratospheric column, as was done by Verhoelst et al. (2021), as the tropospheric column contribution is small, on average 5.1% for the few Pandora observations that include detectable signals of the tropospheric column. We now make this clearer (lines 405-410).

- *Given that the stratosphere is assumed to be fairly uniform over moderate spatial scales, I'm not automatically convinced that even a 1x1 deg model significantly smooths the stratosphere. Dirksen et al. (2011) for example showed that the OMI DOMINO stratospheric product captured a polar vortex event reasonably well by comparison with ground based data.*

A regression slope less than unity suggests that TROPOMI is smoothing spatial gradients. We initially suspected that this is due to the use of the coarser resolution TM5 model smoothing spatial variability in stratospheric $NO_2$, but this could instead be smoothing the spatial variability in the tropopause. We now adjust the text to say more generally that the slope < 1 suggests that higher-resolution features are smoothed by coarser resolution input, such as TM5-MP or 3-hourly meteorology (lines 410-411).

- *One final point I am concerned about is the difference in stratospheric dynamics over Mauna Loa and the rest of the tropics compared with the extratropics due to the Brewer-Dobson circulation. Have you considered the Pandora vs. TROPOMI stratospheric columns at, for example, Fort McKay, Fairbanks, Eureka, or Ny-Alesund to see if this same error in the stratosphere occurs in the mid- and highlatitudes?*

Pandora instruments are total column direct sun photometers, so the only site available that is dominated by contribution from the stratosphere is Mauna Loa. The sites suggested by the reviewer are located in the planetary boundary layer at altitudes ranging from 18 m to 617 m. We use a geometric AMF to calculate the TROPOMI tropospheric columns that would also not be appropriate for these sites, due to enhanced sensitivity to the vertical profile shape of $NO_2$ at these sites. These sites also cover a total column range that is far greater than the sites considered in this work. We find that the maximum coincident midday mean Pandora total column is $7 \times 10^{15}$ molecules $cm^{-2}$ at Eureka, $11 \times 10^{15}$ molecules $cm^{-2}$ at Fairbanks, $19 \times 10^{15}$ molecules $cm^{-2}$ at Fort McKay and $7 \times 10^{15}$ molecules $cm^{-2}$ at Ny-Alesund. Total columns at the free tropospheric sites used in our work are $< 5 \times 10^{15}$ molecules $cm^{-2}$, with the exception of a single point at Altzomoni. We now mention that there are Pandora instruments at remote sites, but that these sample the boundary layer (lines 290-292).

*Line 341 (whole paragraph):*
*1.  Is this really relevant to this paper? It seems focused on general TROPOMI validation, not cloud slicing.*

In retrospect, yes, we agree with the reviewer, as this paragraph summarizes comparisons conducted at low-altitude sites influenced by the boundary layer and using the detailed AMF. We have removed this paragraph.

*2.  If it is relevant, I'm not convinced that the stratospheric variance is the largest reason for these urban/rural differences. Underestimation of satellite NO2 in urban areas and overestimation outside them is a common result of coarse a priori profiles that are too coarse (Russell et al., 2011) because the prior profile is a mix of urban & rural characteristics. Since Pandora is substantially less sensitive to the vertical profile of NO2, it follows that comparisons with Pandora would illuminate this difference.*

We've removed this paragraph, as our extrapolation of results to these other validation studies was erroneous.

*Line 349: "After applying the stratospheric NO2 variance correction" – please explain how this correction is applied.*

We now elaborate on the variance correction (lines 424-426).

*Line 364: "We impose a modest threshold" - does this mean only TROPOMI columns >4e13 are used? TROPOMI columns <4e13 are set to 4e13? Something else?*

We have rewritten "sample" to "only use" so that it is clear we filter out TROPOMI columns $\leq 4 \times 10^{13}$ molecules $cm^{-2}$ (line 472).

*Line 375: "TROPOMI free tropospheric columns (red crosses in Figure 4)" - I am a little confused here. Has this section been discussing TROPOMI observations with just the free tropospheric (excluding boundary layer) or the total tropospheric columns? From Eqs. 1 & 2, there is no indication that the boundary layer has been removed, just the tropospheric VCD was recalculated with a geometry only AMF.*

We have added text to the description of the comparison to ensure it is clear that the comparison of Pandora and TROPOMI at high-altitude sites is effectively a comparison of the total column without boundary layer influence, where the boundary layer typically extends to 1-2 km above the surface and the sites we compare to are at 2.4-4.2 km (lines 278-280). We have also updated the figure caption for Figure 3 for clarity. We do already acknowledge that the Altzomoni site is influenced by pollution from nearby Mexico City (lines 389-390), as evidenced by the greater average contribution of the troposphere (31%) than the other two sites (5-8%) (lines 406-407). Columns there are lower (all but one Pandora total columns <5 $\times 10^{15}$ molecules cm$^{-2}$) than the remote low-altitude Pandora sites (maximum daily means coincident with TROPOMI range from 7 to $19 \times 10^{15}$ molecules cm$^{-2}$).

*Sect 4: UT NO2 retrieval*
*Line 416, paragraph: it is confusing to use "The standalone product version number changes from 01-01-07 to 01-01-08..." as the only transition from describing the FRESCO cloud product to describing the offline cloud product. This is aggravated because, earlier in the paper, FRESCO and ROCINN were used as the names of the products rather than "in the NO2 files" and "the standalone product". Also "These are FRESCO-S from the same data file as TROPOMI NO2 and the standalone offline (OFFL) cloud product," is ambiguous as to whether it means one product is FRESCO-S from the NO2 files and the other is the standalone product or FRESCO-S is the product in both the NO2 and standalone files. This paragraph could be made clearer.*
Thank you for your suggestion. This paragraph has been rewritten and reordered for clarity (lines 553-568).

*Moving the descriptions of the retrievals to the supplement (or removing altogether) could streamline the paper and focus on the cloud slicing aspects.*
- *It seems that the most important difference between these products is whether they treat clouds as reflective surfaces or 3D objects. Focusing this paragraph on that, rather than the details of how these algorithms work, would likely help readability.*
As per the AMT submission guidelines, we are unable to include descriptive text in the supplementary and we are concerned about removing this description altogether, as it provides traceability. Instead, we have shortened the paragraph to include what we view as the most pertinent information (lines 553-568). As a result, we have also removed the sentence discussing the effect of absence of the red band reflectances in TROPOMI on OCRA cloud fractions (line 676).

*Why is there so much more data in the northern high latitudes in Sept-Nov than Jun-Aug (Fig 5)? It seems like the greater summer illumination would make it easier to get data in Jun-Aug. According to Figs. 6 and S2, while there are more clouds in Sep-Nov than Jun-Aug, there are still 1.2 million cloudy pixels in Jun-Aug.*
The high midlatitude coverage by the ROCINN-CAL product is due its greater cloud altitude range than the FRESCO-S cloud top pressure product (Figures 7 and S4) so that the difference

between the min and max cloud top pressures exceeds the 140 hPa difference required in the cloud-slicing steps. We do already discuss this in the text (lines 727-729).

*Line 459: "Coincident gridsquares of the two data sets..." Since the previous sentence was talking about anthropogenic contamination, it isn't immediately clear that the "two data sets" are the two cloud sliced ones. Perhaps a paragraph break would help?*
Thank you for your suggestion. We now include a paragraph break (lines 654-656).

*Line 518: "...as clusters of TROPOMI pixels in the midlatitudes and polar regions overcome the 140 hPa cloud top pressure range..." a bit awkwardly phrased, as "overcome" somewhat implies more clouds exceeding the 140 hPa cutoff, which would imply fewer, not more clusters passing the filter.*
What we have is correct, as only clusters with a cloud top pressure range greater than 140 hPa are used to ensure a dynamic altitude range. This is consistent with Choi et al. (2014) and is already stated in the description of the algorithm for application to the synthetic spectra (line 163).

*Sect 5: Comparison between TROPOMI and OMI*
*Line 531: Was 2005-2007 (rather than a multiyear period overlapping TROPOMI) chosen because that was before the row anomaly began degrading OMI retrievals? If so, would be good to say so to explain why this time period was chosen even though the impact of the 10+ year gap is difficult to quantify.*
That's correct. We now include a sentence stating this (lines 743-744).

*Line 540: Conversely, Romps et al. (2014) suggest lightning will increase 12%/°C. Given that Schumann and Huntrieser published at the beginning of this 10 year gap, a more recent reference would strengthen the claim that lightning has not increased.*
We relied on Schumann and Huntrieser, as this is based on observations rather than model predictions. Most models use the same Price and Rind (1992) relationship between lightning flashes and cloud-top-heights and so obtain a steep increase in lightning flashes with temperature. Other models that follow a different approach using environmental factors such as upward cloud ice flux obtain a less steep increase or a decline in lightning flashes with increases in air temperature due to climate change (Finney et al., 2016; 2018). We now include this discussion in the text (lines 754-757).

*Line 559: "The underestimate in variance in December-February in both products could reflect the need to account for seasonality in the stratospheric variance correction." Cannot tell that the variance is underestimated in both from Fig. 8 since it is just comparing the two satellites' values, without a truth metric. Please indicate how you know both are underestimated.*
We have rewritten this to state that the variance in the TROPOMI products is weaker than that of OMI (line 778).

*Line 560:*
- *"UT NO2 obtained without applying correction factors to TROPOMI stratospheric and tropospheric columns (Figure S5) results in greater data density due to less variance in TROPOMI*

*stratospheric columns, but the discrepancy with OMI is much greater." This raises the question of why TROPOMI needed the 50% reduction in tropospheric columns. Since the TROPOMI:OMI slopes increase, that suggests (assuming that 2005-2007 and 2019-2020 are comparable) that TROPOMI has a multiplicative bias vs. OMI.*

The 50% correction to the tropospheric columns leads to a systematic reduction in cloud-sliced UT NO$_2$, whereas the stratospheric variance correction is relative. Small column densities decrease by a greater proportion than large column densities, with a similar effect on cloud-sliced UT NO$_2$. This is why the slopes in the comparison to OMI exceed unity for both TROPOMI products in all seasons without the correction factors. We have rewritten the sentences that describe the impact of not applying these correction factors to TROPOMI on cloud-sliced UT NO$_2$ (lines 413-414, line 491).

- *Which OMI product (DOMINO v1/v2, NASA SP v1/v2/v3) are you using in this comparison? The QA4ECV product is probably the most similar to the TROPOMI retrieval; in the past, DOMINO has been clearly higher than NASA SP (e.g. Hains et al., 2010), so if the OMI 2005-2007 product uses NASA, then it's possible this differences are retrieval driven (though that comparison is muddled since you're not using the standard tropospheric AMF).*

We now also acknowledge that differences in retrieval approaches could contribute to differences in UT NO$_2$ products compared in Figure 8 and reference both Hains et al. (2010) and the study by Dirksen et al. (2011) (lines 774-776). We also state this in the Conclusions section (lines 824-825).
* * *
**References:**

Brogniez et al., doi:10.1029/2001JD001576, 2002.
Choi et al., doi:10.5194/acp-14-10565-2014, 2014.
Dirksen et al., doi:10.1029/2010JD014943, 2011.
Dubé et al., doi:10.5194/amt-2020-331, 2020.
Finney et al., doi:10.5194/acp-14-12665-2014, 2014.
Finney et al., doi:10.1002/2016gl068825, 2016.
Finney et al., doi:10.1038/s41558-018-0072-6, 2018.
Marais et al., doi:10.5194/acp-18-17017-2018, 2018.
Price and Rind, 10.1029/92JD00719, 1992.
Verhoelst et al., doi:10.5194/amt-14-481-2021, 2021.